ecology, theoretical biology

immune priming, trained immunity, host heterogeneity, *Daphnia magna*, *Pasteuria ramosa*, mathematical modelling

**Authors for correspondence:**
Frida Ben-Ami
e-mail: frida@tauex.tau.ac.il
Roland R. Regoes
e-mail: roland.regoes@env.ethz.ch

# Disentangling non-specific and specific transgenerational immune priming components in host–parasite interactions

Frida Ben-Ami[1], Christian Orlic[2] and Roland R. Regoes[3]

[1]School of Zoology, George S. Wise Faculty of Life Sciences, Tel Aviv University, Tel Aviv 6997801, Israel
[2]Zoologisches Institut, Evolutionsbiologie, Universität Basel, Vesalgasse 1, Basel 4051, Switzerland
[3]Institute of Integrative Biology, ETH Zurich, Zurich 8092, Switzerland

FB-A, 0000-0002-1978-5659; RRR, 0000-0001-8319-5293

Exposure to a pathogen primes many organisms to respond faster or more efficiently to subsequent exposures. Such priming can be non-specific or specific, and has been found to extend across generations. Disentangling and quantifying specific and non-specific effects is essential for understanding the genetic epidemiology of a system. By combining a large infection experiment and mathematical modelling, we disentangle different transgenerational effects in the crustacean model *Daphnia magna* exposed to different strains of the bacterial parasite *Pasteuria ramosa*. In the experiment, we exposed hosts to a high dose of one of three parasite strains, and subsequently challenged their offspring with multiple doses of the same (homologous) or a different (heterologous) strain. We find that exposure of *Daphnia* to *Pasteuria* decreases the susceptibility of their offspring by approximately 50%. This transgenerational protection is not larger for homologous than for heterologous parasite challenges. Methodologically, our work represents an important contribution not only to the analysis of immune priming in ecological systems but also to the experimental assessment of vaccines. We present, for the first time, an inference framework to investigate specific and non-specific effects of immune priming on the susceptibility distribution of hosts—effects that are central to understanding immunity and the effect of vaccines.

## 1. Introduction

Transgenerational effects occur when the phenotype of the parent affects the phenotype of its offspring in addition to the direct effects of the genes contributed by the parent [1–3]. These effects are ubiquitous in nature and have been documented in a wide range of traits and taxa [4–8].

Among the most widely studied transgenerational effects are those involving the transfer of immunity or increased parasite resistance from parents to offspring, commonly found in vertebrates [9,10], but also in invertebrates [11–13]. The latter is particularly intriguing, because until the beginning of this millennium the innate immune system of invertebrates was thought to be capable of only non-specific responses that were unaffected by previous exposures to parasites (e.g. [14]). The potential of innate immune systems to specifically remember previous exposures to pathogens was first supported by phenomenological evidence (reviewed in [15–17]). In particular, it has been shown that invertebrate hosts can be primed against specific parasite species and strains, and the priming effects can extend across life stages and generations [18–22]. There is also growing evidence that the innate immune system of invertebrates shares several homologies with that of vertebrates [16,23,24], although it has been argued that immune memory in invertebrates may be mediated by yet unidentified mechanisms that will not be found by looking for homologies [15,25]. More recently, the interest in potential innate immune memory has been revived, and new

molecular mechanisms are being elucidated in invertebrates [26–28], and even in vertebrates [29].

Immune priming can be non-specific or specific. Specificity here defines the degree to which a primed immune response is able to discriminate among different parasite strains, species, or taxa (e.g. Gram-positive bacteria; [12]). While non-specific immune priming is important for eliciting a general response against a variety of parasites, specific immune priming can provide a targeted, and often more effective and long-lasting protection against reinfections. It is thus crucial to disentangle non-specific from specific immune priming in order to understand which of the two is responsible for an observed immune response.

Studies of transgenerational effects on disease typically subject the parental environment to food stress, for example, shortage of food or food of lower quality [30,31], crowding [32] or challenge them with live, weakened, or heat-killed parasites [33–35]. Thereafter, a variety of traits of the offspring generation are recorded, such as susceptibility to parasites and offspring fecundity, resistance, immunity, and mortality [36–38]. The vast majority of studies on transgenerational effects focused on non-specific immune priming [34,35,39–42]. Only a handful of studies involving invertebrates found evidence for specific transgenerational immune priming. For example, in a serial passage experiment, in which populations of the flour beetle *Tribolium castaneum* were subjected to a regime of challenge with heat-killed and subsequent infection with live *Bacillus thuringiensis* for 11 generations, Khan *et al.* [43] found evidence for the evolution of strain-specific immune priming in the beetles. In another recent study, Norouzitallab *et al.* [44] showed the occurrence of specific immune memory in the brine shrimp *Artemia franciscana*, manifested by increased resistance of the progeny of Vibrio-exposed ancestors towards a homologous bacterial strain when compared with a heterologous strain. Little *et al.* [45] obtained similar results in the crustacean *D. magna* by exposing mothers to one strain of *Pasteuria ramosa* and testing their offspring's fertility following exposure to the same and a different strain.

Because specific immune priming can play an important role in host–parasite interactions at the population level, we combined an experiment with mathematical modelling to disentangle transgenerational effects of non-specific and specific immune priming in *D. magna* and its bacterial parasite *P. ramosa*. We used three isolates of *P. ramosa* to prime mother *Daphnia*, and exposed their offspring to all three isolates in a $3 \times 3$ factorial experiment. This resulted in nine treatment arms: three arms with homologous challenges, and six arms with heterologous challenges.

Instead of exposing host individuals to a single challenge dose of the pathogen, as is done in most studies, we used seven challenge doses ranging over more than five orders of magnitude. We chose multiple challenge doses because this allows us to study not just the average susceptibility but the entire distribution of susceptibilities in the host population under investigation. In particular, this approach can identify if priming affects each host individual uniformly, or if the host response to priming differs across hosts.

The main questions we address are if the susceptibility distribution is affected by priming, and if these potential effects differ for homologous or heterologous challenges. Hereby, we do not only consider priming effects on the mean susceptibility to challenge, but also effects on the variance of susceptibilities across hosts.

# 2. Results

## (a) Dose dependence of infection rates

To determine the existence and extent of these various forms of transgenerational immune priming, we conducted experiments with *D. magna* and three isolates of its parasite *P. ramosa*, P1, P2, and P5. We first exposed genetically identical *Daphnia* to a high dose of one of the three parasite isolates. The exposure lasted 7 days, after which the medium was replaced by parasite-free medium. As a control, a subset of the *Daphnia* were not exposed to any parasite strain. All unexposed control animals remained uninfected throughout the experiment. Overall, this resulted in four treatment groups.

The exposed and control *Daphnia* subsequently produced offspring. All offspring were produced after the exposure of the mothers. We excluded offspring from mothers that did not become infected during the experiment (see electronic supplementary material). We then challenged the offspring individuals from the four treatment groups with seven different doses of each of the three parasite strains (figure 1). Lastly, we assessed the infection status of *Daphnia* offspring 39 days after exposure (on day 44).

The main readout from the experiment is the fraction of infected *Daphnia* as a function of the exposure dose in the homologous and heterologous challenge groups, as well as in the control group (figure 2). Since these data are the result of potentially competing influences of specific and non-specific transgenerational immune priming, a formal method was required to disentangle the effects of maternal exposure on offspring susceptibility.

## (b) Modelling framework

To analyse these data, we extended a mathematical framework we had developed previously. The original framework allowed us to estimate the average infection probability and its inter-individual variance [30,46,47]. The inspiration for our previous work came from frailty mixing models in mathematical epidemiology [48–50], but the approaches are also used in microbial risk assessment [51,52]. The original framework, however, was conceived to analyse infection experiments involving only a single parasite strain and to contrast the susceptibilities of *Daphnia* whose mothers had or had not been exposed to this parasite strain [30,47].

To be able to address the question of non-specific versus specific immune priming in our infection experiment, we extended this framework by incorporating parameters that capture many of the conceivable ways of how exposure of the mothers to a specific parasite strain can alter the susceptibility of the offspring to infection (see electronic supplementary material). This is needed to analyse the results of our fully factorial experiment, in which both the mother and offspring generation were exposed to three parasite strains. In our modelling framework, the baseline susceptibilities of control *Daphnia* to each of these three parameters are denoted by $b_{01}$, $b_{02}$, and $b_{05}$. Hereby, the indices 01, 02, and 05 denote the three control groups of *Daphnia*. The first index '0' signifies that the mothers of the *Daphnia* in these groups were not exposed to a parasite. The second index, '1', '2', or '5', denotes the isolate to which the offspring was exposed, P1, P2, and P5, respectively.

In our extended framework, we separated the potential priming-induced alterations of susceptibility into a heterologous and a homologous component. For example, if the

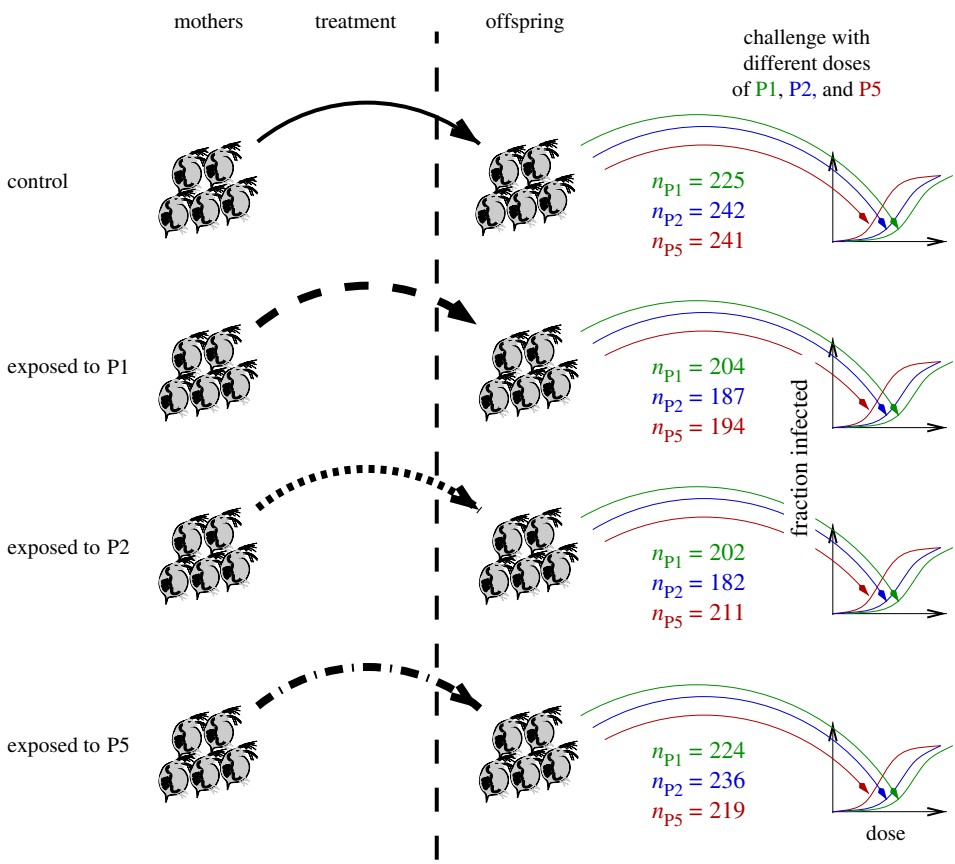

**Figure 1.** Design of our experiment. Mother *Daphnia* were exposed to three different strains of *Pasteuria ramosa*, P1, P2, or P5. A control cohort of mother *Daphnia* was not exposed. The offspring of these mothers were then exposed to seven different challenge doses of P1, P2, or P5. The sample sizes for each group are indicated on the diagram. They amount to approximately 20–30 individuals per strain and challenge dose. In total, we used 2567 individuals. (Online version in colour.)

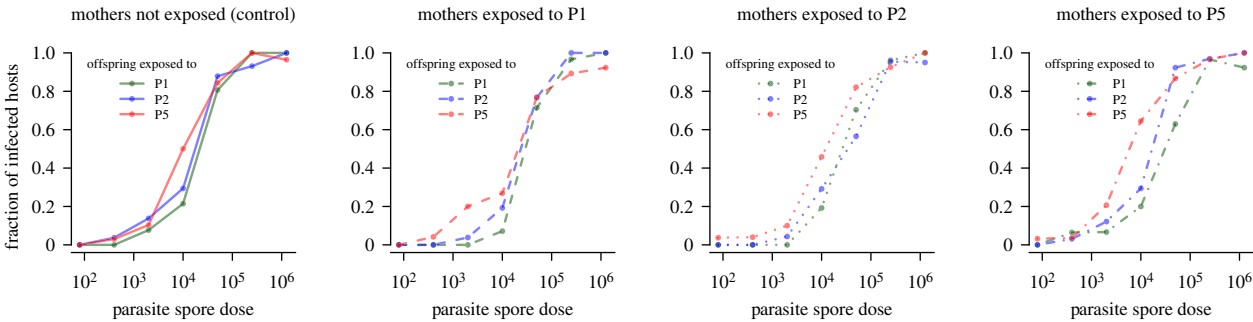

**Figure 2.** Fraction of infected hosts versus parasite challenge dose for each maternal treatment group. The colour and line type scheme is chosen in concordance with the experimental design schematic shown in figure 1. For a figure showing these data by offspring parasite, see electronic supplementary material, figure S2. (Online version in colour.)

mother *Daphnia* was exposed to P1, its offspring may be less susceptible to challenge with any parasite strain. This would constitute non-specific immune priming against heterologous challenge, and is captured by the model parameter $r$ (see electronic supplementary material). Alternatively, maternal exposure to P1 could reduce susceptibility of offspring to P1 specifically, that is, it could partially protect the offspring against homologous challenge. This specific memory effect is captured by the model parameter $m$ (see electronic supplementary material). Our modelling framework also allows for non-specific and specific alterations of the variance of the susceptibility distribution. These effects are captured by the parameters $\rho$ and $\mu$, respectively (see electronic supplementary material).

Our modelling and inference framework has clear advantages over the more commonly applied generalized linear modelling approach. These are detailed in the electronic supplementary material to this paper.

## (c) Baseline susceptibility and heterogeneity estimates

First, we estimated the average susceptibility to each parasite strain $b_{0j}$ and its variance $v_{0j}$ from the infection data of the control group. The baseline susceptibility estimates are an important reference point against which we later tested for non-specific and specific immune priming effects. For P1, P2, and P5, respectively, we obtain $b_{01} = 8.73 \times 10^{-5}$, $b_{02} = 1.68 \times 10^{-4}$, and $b_{05} = 2.49 \times 10^{-4}$ as the average susceptibilities, and $v_{01} = 10^{-9}$, $v_{02} = 0.73$, and $v_{05} = 0.91$ as the susceptibility variances. Figure S4 in the electronic supplementary material shows the likelihoods for the control data. Figure S3A in the electronic supplementary material shows the fits to the control data.

**Table 1.** Model variants considered in the model selection scheme. The highlighted $r - m_i$ model has the strongest statistical support.

| model | description |
|---|---|
| 0 | null model without unspecific or specific immune priming |
| $r$ | unspecific immune priming, equal for each of the three strains |
| $m$ | specific immune priming, equal for each of the three strains |
| $\rho$ | unspecific change in susceptibility variance, equal for each of the three strains |
| $\mu$ | specific change in susceptibility variance, equal for each of the three strains |
| $r - m$ | unspecific and specific immune priming; both effects equal for the three strains |
| $r - \rho$ | unspecific immune priming and unspecific change of the susceptibility variance; both effects equal for the three strains |
| $r - \mu$ | unspecific immune priming and specific change of the susceptibility variance; both effects equal for the three strains |
| $r_i$ | unspecific immune priming; effect may differ among the three strains |
| $r - m_i$ | unspecific and specific immune priming; specific effect may differ among the three strains |
| $r_i - m$ | unspecific and specific immune priming; unspecific effect may differ among the three strains |
| $r - m - \rho$ | unspecific and specific immune priming; unspecific change of susceptibility variance |
| $r - m - \mu$ | unspecific and specific immune priming; specific change of susceptibility variance |
| $r_i - m_i$ | unspecific and specific immune priming; both effects may differ among the three strains |
| $r - m_i - \rho$ | unspecific and specific immune priming; specific effect may differ among the three strains; unspecific change of susceptibility variance |
| $r - m_i - \mu$ | unspecific and specific immune priming; specific effect may differ among the three strains; specific change of susceptibility variance |
| $r - m_i - \rho_i$ | unspecific and specific immune priming; specific effect may differ among the three strains; unspecific change of susceptibility variance that may differ among the three strains |
| $r - m_i - \mu_i$ | unspecific and specific immune priming; specific effect may differ among the three strains; specific change of susceptibility variance that may differ among the three strains |

To study if there is non-specific or specific transgenerational immune priming, we adopted a model selection scheme. We constructed models with increasing complexity, the simplest of which assumes no immune priming effects, and serves as a null model in our analysis. Table 1 lists and defines the models we considered. By fitting the models in order of increasing complexity to our experimental data and comparing the quality of the fits statistically, we test for the existence of non-specific and specific immune priming effects (see electronic supplementary material).

### (d) Evidence for non-specific immune priming

We tested four models that are one step more complex than the null model: $r$, $m$, $\rho$, and $\mu$ (table 1 and figure 3). These model extensions test for the existence of overall immune priming effects of maternal exposure on the average susceptibility or its variation. But this potential effect is assumed not to differ between the parasite strains P1, P2, and P5.

We fitted all four model extensions maximizing the likelihood function described in the electronic supplementary material. While all of these model extensions fit significantly better than the null model, the largest improvement in fit arises from the $r$ model that describes a non-specific, cross-strain immune priming effect (likelihood ratio test: $p = 4.7 \times 10^{-19}$). We estimate an effect $r = 0.43$. This means that maternal exposure, irrespective of the specific parasite strain, reduces the average susceptibility of the offspring to any strain by 43%. This reduction translates into an approximately twofold increase of the $ID_{50}$ (figure 4).

### (e) No evidence for specific immune priming

Because the $r$-model resulted in the largest improvement of model fit we used it as a baseline for the subsequent analysis. We considered four models that are one step more complex than the $r$-model (table 1). Biologically, the most relevant of these are the $r - m$-model and the $r_i$-model. The $r - m$-model allows for specific immune priming in addition to the non-specific effect already described in the $r$-model. This specific effect is captured by the parameter $m$ that denotes the fraction by which the susceptibility to homologous challenge is reduced. The $r_i$-model extends the $r$-model by accommodating potential differences between the non-specific immune priming effects of each parasite strain.

Of the four conceivable models, only the $r - m$-model improves the fit significantly (likelihood ratio test: $p = 0.02$; see also figure 3). Thus, we have evidence for a specific, transgenerational memory of parasite strains. However, the parameter $m$ in this model, which describes how well the maternal parasite strain is remembered, is negative: $m = -0.40$. This can be interpreted as specific facilitation of infection, rather than specific protection, and is thus the opposite of immune priming.

### (f) Maternal exposure to P5 facilitates infection with P5

The $r - m$-model can be extended in various ways (table 1 and figure 3). The most relevant extensions are the $r - m_i$-model and the $r_i - m$-model. The $r - m_i$-model assumes that maternal exposure to any of the three parasite isolates reduces the susceptibility of the offspring non-specifically

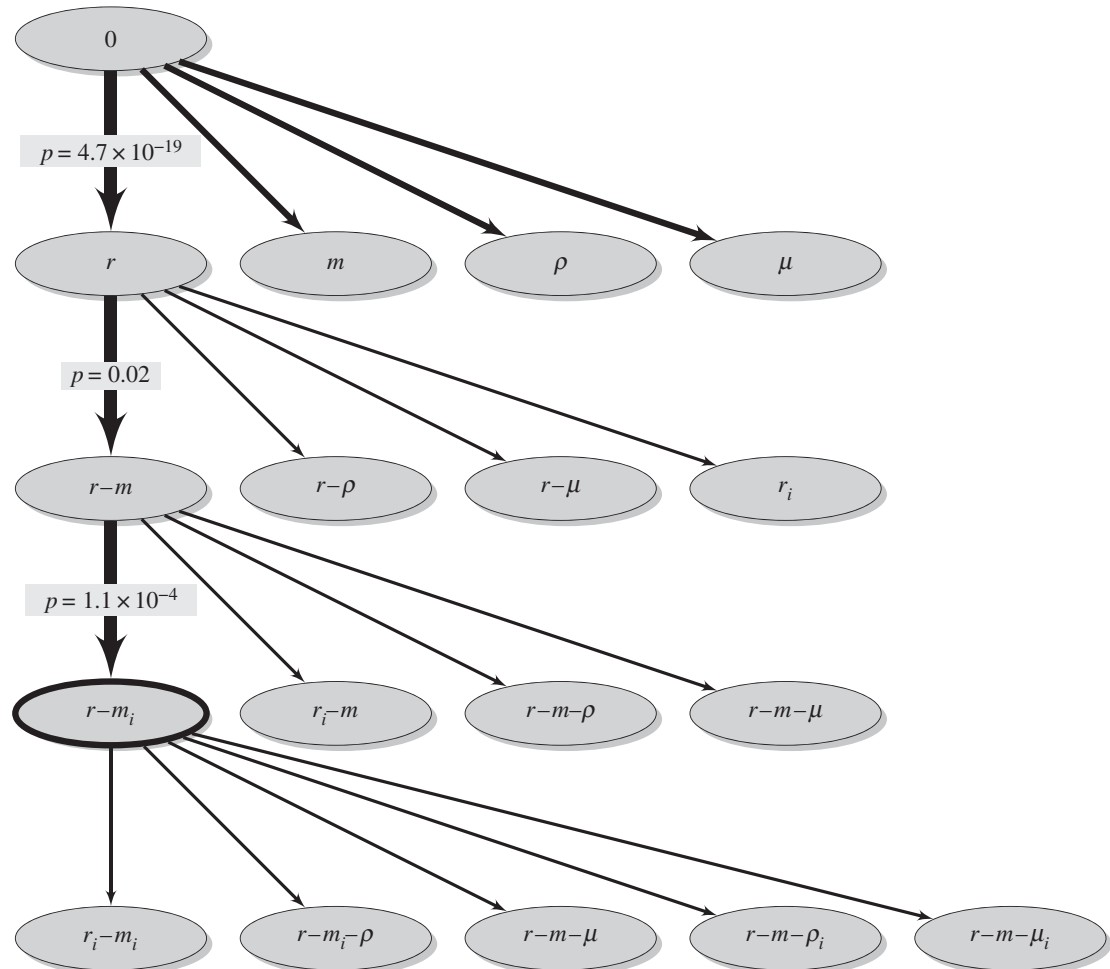

**Figure 3.** Model selection scheme. The thick arrows denote statistically significant model improvements. Statistical significance was determined by a likelihood ratio test between two models. The *p*-values of these tests are shown. The thick ellipse circles the most complex model with statistical support.

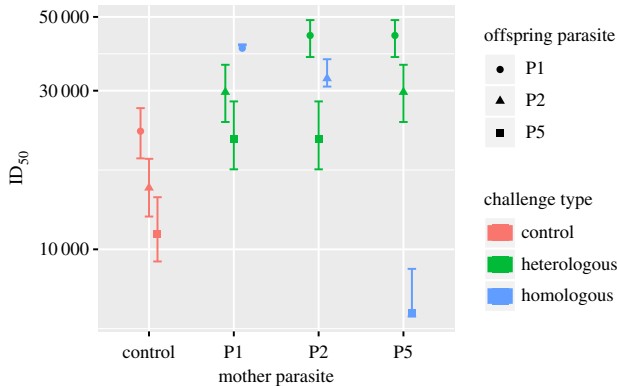

**Figure 4.** $ID_{50}$ by maternal treatment group. Vertical bars show the standard error of the $ID_{50}$ estimates that were calculated by bootstrap (see electronic supplementary material). (Online version in colour.)

by the same fraction $r$. The model further assumes three specific effects, measured by $m_i$, describing how maternal exposure to each parasite isolate reduces the susceptibility of the offspring to homologous challenge with the same parasite isolate. The $r_i - m$-model, by contrast, assumes that the non-specific reduction of the susceptibility of offspring differs for each of the parasite isolates, to which the mothers were exposed. The specific effect, on the other hand, is assumed to be the same for each parasite isolate, that is, the offspring's susceptibility to homologous challenge is assumed to be reduced by the same fraction $m$ for each parasite isolate.

Of all the extensions, we considered, however, only the $r - m_i$-model improves the fit significantly (likelihood ratio test: $p = 1.1 \times 10^{-4}$; see also figure 3). The fit of this model is not improved by any further model extensions (figure 3, bottom row). Hence, the $r - m_i$-model represents the model complex enough to capture all aspects related to non-specific and specific immune priming in the data without over-fitting them. Figure S3B in the electronic supplementary material shows the fits of the best model ($r - m_i$ model) to each group of the data.

The non-specific effect of maternal exposure in the $r - m_i$-model is estimated as $r = 0.48$ with a 95% confidence interval between 0.39 and 0.55. This means that maternal exposure to a parasite isolate reduces the susceptibility of the offspring to any of the three parasite isolates by 48%. The three parameters describing specific immune priming are estimated as $m_1 = -0.054$ (−0.56, 0.29), $m_2 = 0.10$ (−0.50, 0.46), and $m_5 = -2.16$ (−4.07, −0.97). The numbers in parentheses give the 95% confidence intervals. Importantly, only $m_5$ is estimated to be significantly different from 0, and is negative. This means that exposing mothers to P5 facilitates infection with P5. This surprising effect is also reflected in the low $ID_{50}$ estimate for this treatment group (see the lowest point in figure 4).

## 3. Discussion

In the present study, we exposed *D. magna* mothers to three strains of the bacterial parasite *P. ramosa*, and then challenged

the offspring homologously or heterologously. Our aim was to determine if there are transgenerational effects of parasite exposure on the distribution of host susceptibility to infection, and if these are driven by non-specific or specific immune priming. We found strong evidence of non-specific, cross-strain immune priming, which decreases the susceptibility of host offspring to infection by approximately 50%. We found no evidence of specific immune priming that reduces susceptibility to infection with the same strain, with which the mother had been challenged (homologous exposure), when compared with the susceptibility to infection with other strains (heterologous exposure). However, we found that maternal exposure to one particular parasite isolate (P5) facilitates, rather than prevents, offspring infection with this parasite.

A previous study on the same host–parasite system found evidence for strain-specific immune priming [45]. We, by contrast, did not find evidence of specific immune priming fine enough to distinguish among different strains of *Pasteuria*. One important difference between the two studies lies in the trait considered to be affected by maternal parasite exposure: we considered susceptibility to challenge and its variation across individuals in the population, while Little *et al.* [45] focused on offspring fecundity. Furthermore, approximately 1000-fold differences in sensitivities between the two strains used by [45] might indicate a confounding dose effect. Alternatively, specific immune priming might be driven by genotype-by-genotype (GxG) interactions, which are well documented in this system [53], insofar the specific host–parasite combinations would be more likely than others to exhibit specific immune priming. The conflict between our study and that of [45] should not be conflated with the more fundamental criticism voiced against studies of invertebrate immunity [54,55]. As many studies of invertebrate immunity, we did not investigate the molecular basis of the effects we found. Demonstrating increased offspring survival or resistance to parasites following prior parental exposure does not necessarily require the involvement of immunity [56]. But, by relying on the formal concepts and the experimental design principles of population biology, we succeed in elucidating the phenomenological effects of previous exposure on host susceptibility to a yet unsurpassed level of detail.

The mean susceptibility of *Daphnia* to isolate P1 and its variance were significantly lower in comparison with P2 and P5. These results are consistent with earlier studies of P1 and two other *Pasteuria* isolates, P3 and P4, which showed that P1 had the lowest infectivity [47]. Moreover, in a variety of mixed infection scenarios, P1 was found to be more virulent but produced fewer spores than isolates P3/P4 and clone C1 (obtained from isolate P5, [47,57]). If virulent strains produce fewer transmission stages, this could influence the generation of specific immune priming and transgenerational memory.

Our results are consistent with previous studies of transgenerational effects of *Pasteuria* exposure on *Daphnia* susceptibility [30,47]. In those studies, we exposed *Daphnia* to only one *Pasteuria* isolate (P5). Thus, we could not test for specific immune priming. The parameter estimates for this isolate we obtained here however, are inconsistent with those from our previous study [30]. The mean susceptibility of the control group was almost fourfold lower, whereas mean susceptibility of the exposed group was less than twofold lower than previously. Consequently, in the present study, mean susceptibility of the control group was 33%

higher than that of the exposed group, whereas in the previous study, it was 38% lower. While the susceptibility variance of the control group between our two studies are consistent, in the present study exposure to isolate P5 did not lead to the significant increase in the susceptibility variance we found previously.

Since the comparison between the present study and the previous one [30] was unplanned, it is not surprising to find differences between the two studies. Nevertheless, it is important to carefully consider what factors could have led to such a divergence between the studies in the mean and variance of offspring susceptibility in the treatments with control and P5-exposed mothers. First, environmental conditions such as food availability and host density can influence maternal effects [31,32,58]. The duration of exposure could also influence susceptibility [59,60]. However, daily food levels of control and exposed treatments and the duration of exposure of exposed treatments in this study were very similar to those in the previous one [30]. Second, phenotypic heterogeneity in host susceptibility to environmental and physiological factors, such as molecular differences in immune response [61] and within-clone variation in life-history traits (e.g. differences in size at birth; [62]), could also influence the mean and variance of offspring susceptibility. Such heterogeneity would, however, not explain why maternal exposure to parasite isolates P1 or P2 did not facilitate offspring infections as it did for P5. Lastly, the genetic composition of parasite isolate P5 might have changed across studies. Isolates are parasite samples from infected hosts that may contain multiple genotypes [63]. They are a naturally occurring feature of the *Daphnia-Pasteuria* host–parasite system. In the laboratory, isolates are propagated through experimental hosts, to obtain enough spore-carrying cadavers to produce sufficient amounts of spore suspensions. Thus, it might be that over time, some genotypes within the P5 isolate have changed in frequency.

Transgenerational immune priming has been described in a variety of taxa, including insects and crustaceans (reviewed in [56]). Our work adds to the growing literature on transgenerational immune priming in invertebrates. While most studies could not disentangle non-specific from specific immune priming by design because the mothers and the offspring were exposed to the same parasite strain, there is mounting evidence of specific immune priming in invertebrates [33,43–45]. In this study, we present the most extensive dataset and analysis of transgenerational specific immune priming in invertebrates to date. While we find clear evidence for non-specific immune priming across generations, our results on specific immune priming are basically negative. Our evidence for a specific priming effect applies only to one of the three *Pasteria ramosa* isolates (P5) and goes into the 'wrong' direction of facilitation, rather than protection. Our study thus shows the limits of specificity of immune priming in *Daphnia*. According to our findings, *Daphnia* do not inherit a memory of the specific *P. ramosa* isolate to which they were exposed. Our study emphasizes that there are limits of specificity, even in systems where specific immune priming effects have been established. Determining these limits can contribute to identifying the often elusive molecular mechanisms that confer specific priming effects in invertebrates.

The fact that transgenerational immune priming is widespread suggests that this trait has adaptive value. Two

evolutionary hypotheses have been proposed [56]. First, the transfer of immunity to offspring may protect it when it is not yet able to mount its own effective responses. This hypothesis essentially focuses on trade-offs between different life stages (reviewed in [64]). Because we exposed the offspring generation early in life, this hypothesis could, at least in part, be behind the priming effect we have established. Second, if the maternal pathogen environment resembles that of the offspring, immune priming is evolutionary advantageous. Hereby, the exact degree of specificity that is most adaptive depends on how likely it is that mother and offspring are exposed to the same type or strain of pathogen [65,66]. The non-specific priming effects we have found could have evolved in response to persistent pathogen pressure across generations. The fact that we could not find evidence for specific immune priming is consistent with an evolutionary scenario, in which subsequent generations face pressure from various types of pathogens rather, than the same strain of a pathogen, such as *P. ramosa*.

Formally, our work represents an important contribution not only to the analysis of immune priming effects in ecological systems but also to the experimental and epidemiological assessment of vaccines. In the epidemiological setting, frailty models have been used to infer the distribution of susceptibilities and vaccine effects [50]. Most importantly, this line of research gave rise to a more refined perspective on vaccine effects delineated by the two extreme scenarios of leaky and all-or-none effects [48–50,67,68]. A leaky vaccine effect describes a scenario in which the susceptibility of each vaccinated individual is reduced by the same factor. The all-or-none scenario, on the other hand, reflects a vaccine that is 100% effective in a subpopulation of vaccinees, and completely ineffective in the remaining population. These refined concepts of vaccine efficacy have been successfully used to infer the effect of vaccines in the epidemiological setting [49]. In experimental settings, in which the challenge dose and schedule can be better controlled, repeated low-dose challenges or challenges with multiple doses have been used to determine vaccine effects beyond the average reduction of susceptibility [69,70]. Also, the potentially immunizing effect of challenges in repeated schedules has been investigated [71]. However, an extension of these frailty modelling approaches to investigate specific and non-specific effects of immune priming has not been available to date. In this study, we provide such an extension. Furthermore, in our specific host-parasite system the lack of evidence for priming effects on the variance parameters of the susceptibility distributions strongly suggest a predominantly leaky mode of action of the non-specific transgenerational priming on susceptibility.

More generic statistical approaches to analysing the type of experiments we conducted, such as generalized linear models, have a number of shortcomings (see electronic supplementary material). Most importantly, they need to be tweaked to distinguish between homologous and heterologous challenges. Moreover, they do not allow to account for heterogeneity in susceptibility. These aspects, however, are central to understanding immunity and the effect of vaccines because immune memory and vaccines are typically specific to certain pathogen strains. Because vaccines aim to provide specific protection against certain pathogen strains, our inference framework will, therefore, be of use much beyond the example of *D. magna* and their parasites that we presented here.

# 4. Material and methods

## (a) Study organisms

*Daphnia magna* Straus is a cyclical parthenogenetic zooplankton, found in a variety of freshwater habitats, such as ponds and rain pools. In nature, many populations are found to be infected by numerous bacterial, microsporidial, and fungal parasites [72–74]. In the laboratory, clonal lines can be kept for many generations, allowing the exclusion of genetic effects experimentally. One of the most common obligate endoparasites of *D. magna* is the bacterium *P. ramosa* Metchnikoff 1888. This bacterial parasite castrates its host and has a strictly horizontal transmission strategy, by releasing spores from the cadaver of infected *Daphnia* that are ingested by susceptible *Daphnia* [75,76]. The castration, however, is not immediate and hosts can exhibit a burst of reproduction prior to death [76–78].

## (b) Experimental design

Our experimental design is summarized in figure 1. In brief, we initially either exposed *Daphnia* mothers to one of three *P. ramosa* isolates (P1: Gaarzerfeld, Germany, 1997; P2: Kaimes, England, 2002; P5: Moscow, Russia), or left them unexposed as controls. These isolates are parasite samples from infected hosts that may contain multiple genotypes [63]. Isolates are a naturally occurring feature of the *Daphnia-Pasteuria* host–parasite system, and are thus relevant to evolutionary processes in natural populations [76]. Despite the potential genetic heterogeneity of the parasite isolates, specific, heritable interactions with the host have been observed [79].

We exposed the mothers in the exposed groups to the parasite for 7 days. After this time the medium was replaced. The generation of offspring by the mothers occurred after the exposure. Thus, the risk of early exposure of the offspring to the parasite is minimal.

We subsequently collected the offspring of the mothers. The number of offspring per mother ranged from one to seven with a median of two. In the exposed groups, we only included the offspring of mothers that became infected upon exposure (by collecting all offspring and using them in the second-generation experiment, and later discarding from the analysis those hosts whose mothers had not been infected). This was done to ensure that the exposure treatment was as homogeneous as possible. We exposed the offspring to different doses of all three *Pasteuria* isolates, thereby creating one homologous and two heterologous groups per maternal treatment group and dose. We used a single laboratory-maintained *D. magna* clone (HO2 from Hungary) in order to exclude genetic variation among hosts apart from mutations. Isolate P5 was used in a previous study of maternal effects of *D. magna* [30].

For the mothers' generation, we placed 4-day-old juveniles individually in 100 ml jars with 20 ml of artificial medium (ADaM; [80]), and on day 5 all individuals in the exposed treatments were challenged with 50 000 spores of the respective *P. ramosa* isolate. We fed the animals $1 \times 10^6$ algae cells of *Scenedesmus gracilis* per *Daphnia* per day. On day 12, we replaced the medium of all animals with 100 ml of fresh medium, and thereafter changed the medium every week. We increased the food levels on days 6, 9, 11, and 13 to $2 \times 10^6$, $2.5 \times 10^6$, $3 \times 10^6$, and $8 \times 10^6$ algae cells per individual per day, respectively, to accommodate the growing food demand.

For the second generation, we collected offspring daily from the mothers and, at an age of 4 days, offspring were singly placed in 100 ml jars with 20 ml of medium. We assigned the offspring of each mother group randomly to one of seven dose levels (80, 400, 2000, 10 000, 50 000, 250 000, and 1 250 000 spores/animal) or to a control group. On day 5, we exposed all individuals to its respective parasite strain/dose combination, and after a

week, the medium of all animals was replaced with 100 ml of fresh medium. Thereafter, we applied medium replacement and feeding schedules identical to those that we had applied to the mothers. We kept both, mothers and offspring, at $20 \pm 0.5°$ C, and set the light:dark cycle ratio to $16:8$ h. We distributed the jars from all treatment groups randomly across the shelves in a controlled climate room, and rearranged them frequently to prevent position effects.

When offspring individuals died, we recorded the day of their death. The main cause of death of offspring was injuries inflicted when we separated them from their mothers. We assessed individuals that died for infection only if their death occurred more than 16 days after their birth because infection cannot be reliably determined earlier. Animals that died earlier were excluded from the analysis. We ended the experiment on day 44, and scored all animals by eye for infection by examining the colour of infected animals, which lose their typical transparency and turn brownish-red, and also lack eggs. In cases in which we could not unambiguously determine the infection status by eye, we dissected the animal to corroborate infection using a phase-contrast microscope (300–600X).

## (c) Mathematical modelling

While the mathematical modelling and inference framework constitutes a key outcome of our research, and although it is essential to our study, we moved the comprehensive description of the modelling into the electronic supplementary material to comply with the page limit of the Proceedings of the Royal Society B.

Data accessibility. We submitted the experimental data and R-code as electronic supplementary material.

Authors' contributions. F.B.A. and R.R.R. conceived the experiment and the statistical inference framework. F.B.A. and C.O. conducted the experiment. R.R.R. conducted the statistical analysis. F.B.A. and R.R.R. wrote the manuscript.

Competing interests. We declare we have no competing interest.

Funding. F.B.A. gratefully acknowledges the financial support of the Israel Science Foundation (grant no. 938/16).

Acknowledgements. We thank Judith Bouman for discussions on the relationship of our model and commonly used generalized linear models, and Eva Bons for commenting on our manuscript.

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
