## [Reviewer comments · Proceedings of the Royal Society B: Biological Sciences]

Review History

RSPB-2019-0561.R0 (Original submission)

Review form: Reviewer 1

Recommendation

Accept with minor revision (please list in comments)

Scientific importance: Is the manuscript an original and important contribution to its field?

Good

General interest: Is the paper of sufficient general interest?

Good

Quality of the paper: Is the overall quality of the paper suitable?

Good

Is the length of the paper justified?

Yes

Should the paper be seen by a specialist statistical reviewer?

No

Do you have any concerns about statistical analyses in this paper? If so, please specify them explicitly in your report.

No

It is a condition of publication that authors make their supporting data, code and materials available - either as supplementary material or hosted in an external repository. Please rate, if applicable, the supporting data on the following criteria.

Is it accessible?

Yes

Is it clear?

Yes

Is it adequate?

Yes

Do you have any ethical concerns with this paper?

No

Comments to the Author

This is indeed a nice approach to tease apart specific vs. nonspecific immune priming in insects. Overall, I find the study quite relevant to the field, providing novel insights into how survival data could be interpreted using balanced use of empirical and theoretical model. I suggest only a few minor points to be addressed

Introduction

- Line 19: Should it not be 'Exposure of mothers'?
- Line 38-40: Please note that a recent study also showed experimental evolution of strain-specific priming in lab-bred flour beetles (Khan et al. 2017, Proceedings B, doi: 10.1098/rspb.2017.1583.).
- Lines 43-47: It is perhaps more clear to say that the homology lies in innate immune components.
- Lines 84-88: Introduction to vaccine seems abrupt to general readers. Can you please rephrase or put more context to it.

Results & Discussion

- Lines 95-96: Some Daphnia sounds strange. Please change it to 'a subset perhaps'
- Line 117: Can you please clarify the parameters here?
- Lines 120-122 "Hereby, we separated these heterologous and a homologous." The sentence construction seems odd. Can you please rephrase?
- Lines 180-182: Sorry if I missed it: If the potential effect does not vary between P1, P2 & P5, then how are they considering these strains are different.
- Line 185: "We estimate an effect $r = 0.43$." Can you clarify how did you fit the model and get the value of r ?
- Lines 191-192: Different parameters of the r model should be clarified.
- Line 252-253: Could you explain clearly how trade-off between virulence and infectivity influence priming?
- Can the modeling framework be a part of the materials and methods rather than result? This is because this section mostly explains the model parameters.

Methods

- Lines 335-336: "In the exposed groups, we only included the offspring of mothers that became infected upon exposure." How was this ensured?
- Lines 364-365: "The experiment ended on day 44, upon which all animals were scored by eye for infection, and their infection status was recorded." While in the results they mention that "After 30 days, the infection status of each offspring Daphnia was determined, resulting in data

on the fraction of infected *Daphnia* as a function of the exposure dose.”

Review form: Reviewer 2

Recommendation

Major revision is needed (please make suggestions in comments)

Scientific importance: Is the manuscript an original and important contribution to its field?

Excellent

General interest: Is the paper of sufficient general interest?

Good

Quality of the paper: Is the overall quality of the paper suitable?

Poor

Is the length of the paper justified?

No

Should the paper be seen by a specialist statistical reviewer?

No

Do you have any concerns about statistical analyses in this paper? If so, please specify them explicitly in your report.

No

It is a condition of publication that authors make their supporting data, code and materials available - either as supplementary material or hosted in an external repository. Please rate, if applicable, the supporting data on the following criteria.

Is it accessible?

Yes

Is it clear?

Yes

Is it adequate?

Yes

Do you have any ethical concerns with this paper?

No

Comments to the Author

In this manuscript, Ben-Ami and colleagues explore the effect of pathogen priming on transgenerational susceptibility using a fully-factorial experiment and mathematical modeling framework. The experiment they conducted was novel, the findings were generally exciting, and the modeling appeared robust. However, while I found the content generally interesting and the experiment novel, I had a difficult time deciphering the primary findings and felt that the writing needed to be substantially revised to improve clarity, reorganize sections, remove typographical errors, and improve sentence structure. I also felt the work could be improved upon through larger attention to broader theory, and a restructuring for the primary graphs and results to draw attention to the core findings of the work.

General comments:

Line numbers would have been helpful in providing feedback.

There are a few word choices that seem at odds with other literature on similar topics. For example, the authors use "unspecific" throughout when the more frequently used term is "nonspecific".

Introduction:

The introduction could be improved by more clearly stating the broader goal of the study, and some hypotheses the authors are testing with their experiment. While I understand it would be unwieldy to make comprehensive predictions about each prime-challenge combination, a clearer statement of why the study should be conducted and how they will interpret their findings would help direct the reader toward the important results.

Results:

Throughout the results, there are multiple typos and several instances where plural is used when the term should be singular, and vice versa. In addition, many sentences are presented in a choppy manner such that it is difficult to discern the object of subsequent sentences, and the manuscript was much longer than was strictly necessary for understanding the findings.

2.1 Dose dependence

A few specific examples of cases where the writing could be improved:

The authors state:

After 30 days, the infection status of each offspring *Daphnia* was determined, resulting in data on the fraction of infected *Daphnia* as a function of the exposure dose. The experiment is sketched in Figure 1.

Could be written as:

We assessed the infection status of *Daphnia* offspring after 30 days and determined the fraction of infected *Daphnia* as a function of exposure dose.

It would be helpful to move the call out to Figure 1 following the "To determine the existence...", the first sentence of the paragraph. The word "sketch" is informal, and you can just refer to the figure in parentheses after your first sentence about the experiment.

Another example:

"Offspring from mothers that were but did not become infected were not used."

Could be written as:

"We excluded offspring from mothers that did not become infected during the experiment. "

In general, these paragraphs needed to be substantially revised and condensed.

2.2 Modeling framework

The authors need to include the full modeling framework and additional information about how the models were fit in the methods. It is not acceptable to refer a reader to a previous manuscript for all the details.

Referring to the modeling framework as capturing all "conceivable" ways of how exposure could alter susceptibility is a little strange. I would recommend revising this. Again, choppy sentences make the description a bit difficult to understand and there are some awkward switches between active and passive voice.

In the model description, there is no explanation of the parameters following the description of the model. This needs to be added. It is also unclear whether each b here is an individual *Daphnia*, and the subscripts need to be described.

From "The above modeling framework..." to end of the section - it is unclear why any of this

information is in the results section, particularly in the context of all the missing information about the model. This seems like a response to reviewers that is oddly placed in the results section but contains no references, and I had no idea why the authors were explaining this. This would be better suited to another paper or explained in the supplemental material.

The modeling description is insufficient to understand how the authors estimated parameters or fit the model. I would recommend adding all the details to the Methods section with a brief introductory description here.

I found Figure 3 to be confusing and was not sure that it helped with interpretation. I do not think this needs to be included as a main figure, or it needs to be substantially revised to explain how it helps to assess whether immunity is specific or non-specific.

2.3 Baseline susceptibility

It is unclear why information gained on the estimates of average and baseline susceptibility is important and therefore why this is being described. The last paragraph of this section needs to be revised for grammar and writing.

2.4 Evidence for..

"There are four" - Should be "We tested four possible models"

"While all model extensions".. This is a central result but is lost by having to compare ID50 values between 5A and 5B. It would be better to include a summary graph showing the ID50s and various fits, or some other summary figure that could better highlight this result.

2.4 & 2.5 - These headers seem to be indicating identical results

2.6. last paragraph: The way these parameters are described makes it very difficult to understand what they are indicating. They seem to suggest facilitation of infection for the P5 strain, but this was not clear.

Discussion:

In general, I felt the discussion was fairly narrow and lacked a broader view of what the results indicated about transgenerational priming as it might apply to other systems. Also, the writing needs to be carefully revised to be clearer and ensure all sentences are complete.

"The conflict between our study" ..to the end of the paragraph - This was an odd ending to this paragraph, and I did not think it was necessary.

"In summary," - this section to the end is the strongest part of the discussion. I think integrating this broader more theoretical context into the specific findings would improve the discussion.

Methods

In general, the authors switch between active "We did this" voice and passive "The food levels were this" voice. This should be revised.

"Offspring died due to injury inflicted"... This is awkwardly written and unclear why this happened. In general, the writing of this paragraph should be significantly improved.

Figure 1. This is a very helpful conceptual figure. I really liked this diagram.

Figure 2. The line thickness made this very difficult to discern. I would recommend changing to color instead of line thickness. Also, the abbreviations are unnecessary and the description can be written in full. I also was uncertain why any of these were displayed and what quantitative result

was supposed to be obtained from this figure. The authors state that more formal analysis is needed to understand these, so why are they being shown in the main text at all?

Figure 3. As mentioned above, I did not understand this figure nor could I discern how the authors came up with these predictions. More information is necessary here.

Figure 4: Does this figure really need to be included in the main text? It doesn't provide much information in isolation (we need to refer to Table 1 to understand the models) and the authors state in the results which models were most supported.

Figure 5. Aside from Figure 2, which the authors state does not provide clear insight into their question, this is the primary data figure in the manuscript. However, the display of these graphs makes it very difficult to compare among groups and generally understand what each of these different panels indicates. The use of green color makes it very difficult to see, and the dotted versus dashed line is unnecessary, particularly when plotted on different panels. It would also be easier to compare across groups if these were plotted on the same graph, for example combining either columns or rows. Including a separate panel of ID50s (for example as a bar graph) would also provide for insight than comparing numbers across graphs.

Decision letter (RSPB-2019-0561.R0)

23-Apr-2019

Dear Dr Regoes,

I am writing to inform you that your manuscript RSPB-2019-0561 entitled "Disentangling unspecific and specific transgenerational immune priming components in host-parasite interactions" has, in its current form, been rejected for publication in Proceedings B.

This action has been taken on the advice of referees, who have recommended that substantial revisions are necessary. With this in mind we would be happy to consider a resubmission, provided the comments of the referees are fully addressed. However please note that this is not a provisional acceptance.

To upload a resubmitted manuscript, log into <http://mc.manuscriptcentral.com/prsb> and enter your Author Centre, where you will find your manuscript title listed under "Manuscripts with

Decisions." Under "Actions," click on "Create a Resubmission." Please be sure to indicate in your cover letter that it is a resubmission, and supply the previous reference number.

Sincerely,
 Proceedings B
 mailto: proceedingsb@royalsociety.org

Associate Editor

Comments to Author:

The manuscript 'Disentangling unspecific and specific transgenerational immune priming components in host-parasite interactions' was reviewed by myself and two experts in the field. As you can see from their reviews, the reviewers both thought the manuscript tested important concepts and would be of broad interest to readers. However, there are some significant points and clarifications that were raised and that should be addressed prior to publication including the following: 1) there are many writing errors throughout that make it difficult to interpret the findings. This paper needs to be carefully edited in its entirety; 2) both reviewers emphasize the need for more details about the modeling and model fit. The authors need to include the full modeling framework in this manuscript rather than citing a previous paper; 3) the discussion should be broadened to address importance of these findings across the field and other systems as well; 4) the figures need to be reworked to better convey the main important findings of the paper. In addition to these issues, there is a long list of specific suggested changes the authors should address as well.

Reviewer(s)' Comments to Author:

Referee: 1

Comments to the Author(s)

This is indeed a nice approach to tease apart specific vs. nonspecific immune priming in insects. Overall, I find the study quite relevant to the field, providing novel insights into how survival data could be interpreted using balanced use of empirical and theoretical model.

I suggest only a few minor points to be addressed

Introduction

- Line 19: Should it not be 'Exposure of mothers'?
- Line 38-40: Please note that a recent study also showed experimental evolution of strain-specific priming in lab-bred flour beetles (Khan et al. 2017, Proceedings B, doi: 10.1098/rspb.2017.1583.).
- Lines 43-47: It is perhaps more clear to say that the homology lies in innate immune components.
- Lines 84-88: Introduction to vaccine seems abrupt to general readers. Can you please rephrase or put more context to it.

Results & Discussion

- Lines 95-96: Some Daphnia sounds strange. Please change it to 'a subset perhaps'
- Line 117: Can you please clarify the parameters here?
- Lines 120-122 "Hereby, we separated these heterologous and a homologous." The sentence construction seems odd. Can you please rephrase?
- Lines 180-182: Sorry if I missed it: If the potential effect does not vary between P1, P2 & P5, then how are they considering these strains are different.
- Line 185: "We estimate an effect $r = 0.43$." Can you clarify how did you fit the model and get the value of r ?
- Lines 191-192: Different parameters of the r model should be clarified.
- Line 252-253: Could you explain clearly how trade-off between virulence and infectivity influence priming?

- Can the modeling framework be a part of the materials and methods rather than result? This is because this section mostly explains the model parameters.

Methods

- Lines 335-336: "In the exposed groups, we only included the offspring of mothers that became infected upon exposure." How was this ensured?

- Lines 364-365: "The experiment ended on day 44, upon which all animals were scored by eye for infection, and their infection status was recorded." While in the results they mention that "After 30 days, the infection status of each offspring Daphnia was determined, resulting in data on the fraction of infected Daphnia as a function of the exposure dose."

Referee: 2

Comments to the Author(s)

In this manuscript, Ben-Ami and colleagues explore the effect of pathogen priming on transgenerational susceptibility using a fully-factorial experiment and mathematical modeling framework. The experiment they conducted was novel, the findings were generally exciting, and the modeling appeared robust. However, while I found the content generally interesting and the experiment novel, I had a difficult time deciphering the primary findings and felt that the writing needed to be substantially revised to improve clarity, reorganize sections, remove typographical errors, and improve sentence structure. I also felt the work could be improved upon through larger attention to broader theory, and a restructuring for the primary graphs and results to draw attention to the core findings of the work.

General comments:

Line numbers would have been helpful in providing feedback.

There are a few word choices that seem at odds with other literature on similar topics. For example, the authors use "unspecific" throughout when the more frequently used term is "nonspecific".

Introduction:

The introduction could be improved by more clearly stating the broader goal of the study, and some hypotheses the authors are testing with their experiment. While I understand it would be unwieldy to make comprehensive predictions about each prime-challenge combination, a clearer statement of why the study should be conducted and how they will interpret their findings would help direct the reader toward the important results.

Results:

Throughout the results, there are multiple typos and several instances where plural is used when the term should be singular, and vice versa. In addition, many sentences are presented in a choppy manner such that it is difficult to discern the object of subsequent sentences, and the manuscript was much longer than was strictly necessary for understanding the findings.

2.1 Dose dependence

A few specific examples of cases where the writing could be improved:

The authors state:

After 30 days, the infection status of each offspring Daphnia was determined, resulting in data on the fraction of infected Daphnia as a function of the exposure dose. The experiment is sketched in Figure 1.

Could be written as:

We assessed the infection status of Daphnia offspring after 30 days and determined the fraction of infected Daphnia as a function of exposure dose.

It would be helpful to move the call out to Figure 1 following the "To determine the existence...", the first sentence of the paragraph. The word "sketch" is informal, and you can just refer to the figure in parentheses after your first sentence about the experiment.

Another example:

"Offspring from mothers that were but did not become infected were not used."

Could be written as:

"We excluded offspring from mothers that did not become infected during the experiment. "

In general, these paragraphs needed to be substantially revised and condensed.

2.2 Modeling framework

The authors need to include the full modeling framework and additional information about how the models were fit in the methods. It is not acceptable to refer a reader to a previous manuscript for all the details.

Referring to the modeling framework as capturing all "conceivable" ways of how exposure could alter susceptibility is a little strange. I would recommend revising this. Again, choppy sentences make the description a bit difficult to understand and there are some awkward switches between active and passive voice.

In the model description, there is no explanation of the parameters following the description of the model. This needs to be added. It is also unclear whether each b here is an individual *Daphnia*, and the subscripts need to be described.

From "The above modeling framework..." to end of the section - it is unclear why any of this information is in the results section, particularly in the context of all the missing information about the model. This seems like a response to reviewers that is oddly placed in the results section but contains no references, and I had no idea why the authors were explaining this. This would be better suited to another paper or explained in the supplemental material.

The modeling description is insufficient to understand how the authors estimated parameters or fit the model. I would recommend adding all the details to the Methods section with a brief introductory description here.

I found Figure 3 to be confusing and was not sure that it helped with interpretation. I do not think this needs to be included as a main figure, or it needs to be substantially revised to explain how it helps to assess whether immunity is specific or non-specific.

2.3 Baseline susceptibility

It is unclear why information gained on the estimates of average and baseline susceptibility is important and therefore why this is being described. The last paragraph of this section needs to be revised for grammar and writing.

2.4 Evidence for..

"There are four" - Should be "We tested four possible models"

"While all model extensions".. This is a central result but is lost by having to compare ID50 values between 5A and 5B. It would be better to include a summary graph showing the ID50s and various fits, or some other summary figure that could better highlight this result.

2.4 & 2.5 - These headers seem to be indicating identical results

2.6. last paragraph: The way these parameters are described makes it very difficult to understand what they are indicating. They seem to suggest facilitation of infection for the P5 strain, but this was not clear.

Discussion:

In general, I felt the discussion was fairly narrow and lacked a broader view of what the results indicated about transgenerational priming as it might apply to other systems. Also, the writing needs to be carefully revised to be clearer and ensure all sentences are complete.

"The conflict between our study" ..to the end of the paragraph - This was an odd ending to this paragraph, and I did not think it was necessary.

"In summary," - this section to the end is the strongest part of the discussion. I think integrating this broader more theoretical context into the specific findings would improve the discussion.

Methods

In general, the authors switch between active "We did this" voice and passive "The food levels were this" voice. This should be revised.

"Offspring died due to injury inflicted"... This is awkwardly written and unclear why this happened. In general, the writing of this paragraph should be significantly improved.

Figure 1. This is a very helpful conceptual figure. I really liked this diagram.

Figure 2. The line thickness made this very difficult to discern. I would recommend changing to color instead of line thickness. Also, the abbreviations are unnecessary and the description can be written in full. I also was uncertain why any of these were displayed and what quantitative result was supposed to be obtained from this figure. The authors state that more formal analysis is needed to understand these, so why are they being shown in the main text at all?

Figure 3. As mentioned above, I did not understand this figure nor could I discern how the authors came up with these predictions. More information is necessary here.

Figure 4: Does this figure really need to be included in the main text? It doesn't provide much information in isolation (we need to refer to Table 1 to understand the models) and the authors state in the results which models were most supported.

Figure 5. Aside from Figure 2, which the authors state does not provide clear insight into their question, this is the primary data figure in the manuscript. However, the display of these graphs makes it very difficult to compare among groups and generally understand what each of these different panels indicates. The use of green color makes it very difficult to see, and the dotted versus dashed line is unnecessary, particularly when plotted on different panels. It would also be easier to compare across groups if these were plotted on the same graph, for example combining either columns or rows. Including a separate panel of ID50s (for example as a bar graph) would also provide for insight than comparing numbers across graphs.

Author's Response to Decision Letter for (RSPB-2019-0561.R0)

See Appendix A.

RSPB-2019-2386.R0 (Revision)

Review form: Reviewer 3

Recommendation

Accept as is

Scientific importance: Is the manuscript an original and important contribution to its field?

Excellent

General interest: Is the paper of sufficient general interest?

Good

Quality of the paper: Is the overall quality of the paper suitable?

Excellent

Is the length of the paper justified?

Yes

Should the paper be seen by a specialist statistical reviewer?

No

Do you have any concerns about statistical analyses in this paper? If so, please specify them explicitly in your report.

No

It is a condition of publication that authors make their supporting data, code and materials available - either as supplementary material or hosted in an external repository. Please rate, if applicable, the supporting data on the following criteria.

Is it accessible?

Yes

Is it clear?

N/A

Is it adequate?

N/A

Do you have any ethical concerns with this paper?

No

Comments to the Author

This manuscript describes a set of fully factorial trans-generational priming experiments using three isolates of *P. ramosa* in *Daphnia*. Notably, the experiments include full dose-response estimates of susceptibility in the offspring to derive not only the mean changes in susceptibility, but also the variance in phenotypes over exposure dose. The manuscript further describes a mathematical framework that can be applied to this data to isolate interpretable estimates for the contributions of specific and non-specific priming/facilitation of a primary exposure upon the susceptibility of a host to a secondary exposure.

I really enjoyed this manuscript and think the framework ought to be a model for susceptibility and priming studies going forward. In fact, I can think of a couple of recently published specificity in priming studies that would have been further improved by this approach (there is a

new and directly relevant one that the authors might be interested in citing – Ferro et al. 2019 PNAS 116(41)). I thought the discussion about the advantages of this modeling framework over GLMs was particularly thoughtful, and should be appealing to those who might not otherwise appreciate the advantages of this approach.

I think the ESM plays a role of outsized importance in getting the big picture across here, but I also don't really have suggestions about how to better integrate some of the details into the main manuscript without significantly altering the length. The main text itself does a good job of pointing readers to the ESM, so hopefully it will be read just as thoroughly as the main text.

I think the manuscript is well written and clearly communicates the key concepts. I have no suggestions for revisions that would significantly improve the manuscript at this point.

Review form: Reviewer 4

Recommendation

Major revision is needed (please make suggestions in comments)

Scientific importance: Is the manuscript an original and important contribution to its field?

Good

General interest: Is the paper of sufficient general interest?

Good

Quality of the paper: Is the overall quality of the paper suitable?

Good

Is the length of the paper justified?

Yes

Should the paper be seen by a specialist statistical reviewer?

No

Do you have any concerns about statistical analyses in this paper? If so, please specify them explicitly in your report.

No

It is a condition of publication that authors make their supporting data, code and materials available - either as supplementary material or hosted in an external repository. Please rate, if applicable, the supporting data on the following criteria.

Is it accessible?

Yes

Is it clear?

Yes

Is it adequate?

Yes

Do you have any ethical concerns with this paper?

No

Comments to the Author

This manuscript examines specific versus non-specific transgenerational immune priming in a *Daphnia*-bacterial parasite system, using distinct parasite strains. This is clearly a very well done and thorough study, which uses a novel and powerful modeling framework with strong broader utility well beyond this system. While the methodological advances of the paper are clear from reading it, I found that the discussion did not clearly elucidate the main take-home points of the study for our understanding of transgenerational immune priming more broadly. The lack of evidence for specific immune memory is interesting, especially given prior studies showing its presence (in one case even on the same system). But what can we conclude from this? It is interesting that most of the invertebrate studies testing for specific immunity focus on strain-level specificity of immune priming, but perhaps there is specificity present more frequently at higher levels of biological organization (bacterial species, if one can even define such a thing!). I found the use of distinct doses for the challenge infection really interesting (as well as the lack of detectable effects of priming on variance), but the authors never really highlight anything about these results in the discussion (other than mentioning that their models allow for looking at heterogeneity). Thus, I was left wanting a bit in terms of what we can conclude about this field more generally after such a thorough and well-done study. The “facilitation” result was also very strange and difficult to explain in light of prior results. However, I know that ecology is messy, and sometimes there is no good explanation. Below are my specific comments, which I hope will be helpful for any revisions:

Line 19-20: awkwardly worded – can you say “with multiple doses of the same (homologous) or a different (heterologous) strain.”

Line 23: “experimental assessment of vaccines” seems a bit strong- I might clarify that you are talking about your modeling framework here (the experimental assessment wording implies otherwise, in my opinion)

Lines 108: I think you should refer to supplemental materials here when you mention that you excluded offspring from moms that did not become infected.

Lines 110-111: rather than saying the experiment is diagrammed in Figure 1, just cite Fig 1 after one of the prior sentences.

Lines 113-114: similar to the above, just cite Figure 2 at the end of the prior sentence rather than having a sentence indicating what the figure shows.

Lines 125-126: awkward wording. Perhaps replace with “contrast the susceptibilities of *Daphnia* whose mothers had or had not been exposed to this parasite strain.”

Line 171: “strain” should be plural here

Lines 210-211: again, just cite the figure in the earlier sentence

Line 220: simply say “(Figure 4)”

Line 233: specify that this was in the same system, as that is a critical detail

Lines 264-266: this all seems unnecessary to write out – why not just say the later part “in the present study mean susceptibility of the control group was almost fourfold lower than previously...” rather than writing out all the estimates from each group, which is tedious to read.

Lines 271-273: awkward sentence. Again, I don’t think you need to give actual variance values- just give the percentage decrease.

Line 304: this paragraph ends with no conclusion – what can we conclude from your study and how does that change our understanding of specific immune priming in *Daphnia*, or more broadly?

Lines 305-320: while this is important (and I think added in response to prior reviewers), it's not explicitly connected at all to your study and thus it sounds very out of place. Can you link it to what you found regarding the presence of nonspecific but not specific immune priming? In what cases would one be more beneficial than the other?

Lines 321-338: I like that you discuss where your modeling framework came from but you don't discuss any of the heterogeneity results of your study, despite the unique use of multiple doses. Is it surprising that you didn't find any change in variance with priming? Does this shed light on whether transgenerational immune priming acts akin to all-or-none versus leaky vaccine effects? I assume leaky given that there was no variance effect, but I haven't thought very deeply about this and you all have. It just seems strange that you take the time to describe these hypotheses but don't relate them at all to your own study. And given the use of the varying doses, which is quite unique, it would be nice to highlight some of the inference you can gain from that aspect of the study.

Line 341: this seems like an overstatement. Generalized linear models can certainly distinguish between homologous and heterologous challenges- though not as well as your method.

Line 356: this may be a naïve question but how is transgenerational immune priming relevant in a castrating parasite? I assume the offspring are produced just prior to castration?

Line 404: a bit more details on how you score by eye would be helpful.

Table 1. Can you bold the model that you found the strongest evidence for, to make it easy for the reader?

Fig 2: I agree with prior reviewers that this figure is not particularly useful, especially with the addition of Fig 4 (could confidence intervals of some kind be added to that figure?). If you do keep in Fig 2, I would recommend adding the control lines (unprimed mothers) to each of the other three strains for comparison and remove the first panel altogether- otherwise it is impossible to conclude anything from the graphs now that they are separate (though I understand why they were separated out for visual clarity...)

Fig 3: Is it possible to combine this with the Table in some way?

Fig 4: as mentioned in comment above, can you add binomial confidence intervals of some kind to these?

Decision letter (RSPB-2019-2386.R0)

5 December 2019

Dear Dr Regoes:

Your manuscript has now been peer reviewed and the reviews have been assessed by an Associate Editor. The reviewers' comments (not including confidential comments to the Editor) and the comments from the Associate Editor are included at the end of this email for your

reference. As you will see, the reviewers and the Editors have raised some concerns with your manuscript and we would like to invite you to revise your manuscript to address them.

We very much appreciate the significant improvement in your revised MS, but there remains some important suggestions from referee # 4, that I endorse. As you know, at the PRSB we always aim to publish articles that advance the field significantly with findings of broad generic interest. There is scope within your MS to enhance these elements further, with one final opportunity. While the paper does move the field forward from a methodological perspective, it does not sufficiently make clear how the results move forward our understanding of transgenerational immune priming conceptually. While the use of multiple doses in your design is commended, additional consideration of your findings from the variance / heterogeneity perspective, would be especially informative. Please note, as you will see below, it is only in exceptional circumstances do we allow multiple rounds of revision. While there is a collective view that your work has significant potential, we do require additional revision. I would hope that you are able to respond constructively and fully, since as with all peer review processes, the invitation to revise your MS does not carry with it any assumption of eventual publication. Nevertheless, we would welcome the opportunity to consider your MS one final time.

Research ethics:

Use of animals and field studies:

If you wish to submit your data to Dryad (<http://datadryad.org/>) and have not already done so you can submit your data via this link [http://datadryad.org/submit?journalID=RSPB&manu=\(Document not available\)](http://datadryad.org/submit?journalID=RSPB&manu=(Document%20not%20available)), which will take you to your unique entry in the Dryad repository.

Please submit a copy of your revised paper within three weeks. If we do not hear from you within this time your manuscript will be rejected. If you are unable to meet this deadline please let us know as soon as possible, as we may be able to grant a short extension.

Best wishes,
Professor Gary Carvalho
mailto:proceedingsb@royalsociety.org

Associate Editor
Comments to Author:
See additional Editor's comments above.

Reviewer(s)' Comments to Author:

Referee: 3

Comments to the Author(s).

This manuscript describes a set of fully factorial trans-generational priming experiments using three isolates of *P. ramosa* in *Daphnia*. Notably, the experiments include full dose-response

estimates of susceptibility in the offspring to derive not only the mean changes in susceptibility, but also the variance in phenotypes over exposure dose. The manuscript further describes a mathematical framework that can be applied to this data to isolate interpretable estimates for the contributions of specific and non-specific priming/facilitation of a primary exposure upon the susceptibility of a host to a secondary exposure.

I really enjoyed this manuscript and think the framework ought to be a model for susceptibility and priming studies going forward. In fact, I can think of a couple of recently published specificity in priming studies that would have been further improved by this approach (there is a new and directly relevant one that the authors might be interested in citing – Ferro et al. 2019 PNAS 116(41)). I thought the discussion about the advantages of this modeling framework over GLMs was particularly thoughtful, and should be appealing to those who might not otherwise appreciate the advantages of this approach.

I think the ESM plays a role of outsized importance in getting the big picture across here, but I also don't really have suggestions about how to better integrate some of the details into the main manuscript without significantly altering the length. The main text itself does a good job of pointing readers to the ESM, so hopefully it will be read just as thoroughly as the main text.

I think the manuscript is well written and clearly communicates the key concepts. I have no suggestions for revisions that would significantly improve the manuscript at this point.

Referee: 4

Comments to the Author(s).

This manuscript examines specific versus non-specific transgenerational immune priming in a *Daphnia*-bacterial parasite system, using distinct parasite strains. This is clearly a very well done and thorough study, which uses a novel and powerful modeling framework with strong broader utility well beyond this system. While the methodological advances of the paper are clear from reading it, I found that the discussion did not clearly elucidate the main take-home points of the study for our understanding of transgenerational immune priming more broadly. The lack of evidence for specific immune memory is interesting, especially given prior studies showing its presence (in one case even on the same system). But what can we conclude from this? It is interesting that most of the invertebrate studies testing for specific immunity focus on strain-level specificity of immune priming, but perhaps there is specificity present more frequently at higher levels of biological organization (bacterial species, if one can even define such a thing!). I found the use of distinct doses for the challenge infection really interesting (as well as the lack of detectable effects of priming on variance), but the authors never really highlight anything about these results in the discussion (other than mentioning that their models allow for looking at heterogeneity). Thus, I was left wanting a bit in terms of what we can conclude about this field more generally after such a thorough and well-done study. The "facilitation" result was also very strange and difficult to explain in light of prior results. However, I know that ecology is messy, and sometimes there is no good explanation. Below are my specific comments, which I hope will be helpful for any revisions:

Line 19-20: awkwardly worded – can you say “with multiple doses of the same (homologous) or a different (heterologous) strain.”

Line 23: “experimental assessment of vaccines” seems a bit strong- I might clarify that you are talking about your modeling framework here (the experimental assessment wording implies otherwise, in my opinion)

Lines 108: I think you should refer to supplemental materials here when you mention that you excluded offspring from moms that did not become infected.

Lines 110-111: rather than saying the experiment is diagramed in Figure 1, just cite Fig 1 after one of the prior sentences.

Lines 113-114: similar to the above, just cite Figure 2 at the end of the prior sentence rather than having a sentence indicating what the figure shows.

Lines 125-126: awkward wording. Perhaps replace with “contrast the susceptibilities of *Daphnia* whose mothers had or had not been exposed to this parasite strain.”

Line 171: “strain” should be plural here

Lines 210-211: again, just cite the figure in the earlier sentence

Line 220: simply say “(Figure 4)”

Line 233: specify that this was in the same system, as that is a critical detail

Lines 264-266: this all seems unnecessary to write out – why not just say the later part “in the present study mean susceptibility of the control group was almost fourfold lower than previously...” rather than writing out all the estimates from each group, which is tedious to read.

Lines 271-273: awkward sentence. Again, I don’t think you need to give actual variance values- just give the percentage decrease.

Line 304: this paragraph ends with no conclusion – what can we conclude from your study and how does that change our understanding of specific immune priming in *Daphnia*, or more broadly?

Lines 305-320: while this is important (and I think added in response to prior reviewers), it’s not explicitly connected at all to your study and thus it sounds very out of place. Can you link it to what you found regarding the presence of nonspecific but not specific immune priming? In what cases would one be more beneficial than the other?

Lines 321-338: I like that you discuss where your modeling framework came from but you don’t discuss any of the heterogeneity results of your study, despite the unique use of multiple doses. Is it surprising that you didn’t find any change in variance with priming? Does this shed light on whether transgenerational immune priming acts akin to all-or-none versus leaky vaccine effects? I assume leaky given that there was no variance effect, but I haven’t thought very deeply about this and you all have. It just seems strange that you take the time to describe these hypotheses but don’t relate them at all to your own study. And given the use of the varying doses, which is quite unique, it would be nice to highlight some of the inference you can gain from that aspect of the study.

Line 341: this seems like an overstatement. Generalized linear models can certainly distinguish between homologous and heterologous challenges- though not as well as your method.

Line 356: this may be a naïve question but how is transgenerational immune priming relevant in a castrating parasite? I assume the offspring are produced just prior to castration?

Line 404: a bit more details on how you score by eye would be helpful.

Table 1. Can you bold the model that you found the strongest evidence for, to make it easy for the reader?

Fig 2: I agree with prior reviewers that this figure is not particularly useful, especially with the

addition of Fig 4 (could confidence intervals of some kind be added to that figure?). If you do keep in Fig 2, I would recommend adding the control lines (unprimed mothers) to each of the other three strains for comparison and remove the first panel altogether- otherwise it is impossible to conclude anything from the graphs now that they are separate (though I understand why they were separated out for visual clarity...)

Fig 3: Is it possible to combine this with the Table in some way?

Fig 4: as mentioned in comment above, can you add binomial confidence intervals of some kind to these?

Author's Response to Decision Letter for (RSPB-2019-2386.R0)

See Appendix B.

RSPB-2019-2386.R1 (Revision)

Review form: Reviewer 1

Recommendation

Accept as is

Scientific importance: Is the manuscript an original and important contribution to its field?

Excellent

General interest: Is the paper of sufficient general interest?

Good

Quality of the paper: Is the overall quality of the paper suitable?

Good

Is the length of the paper justified?

Yes

Should the paper be seen by a specialist statistical reviewer?

No

Do you have any concerns about statistical analyses in this paper? If so, please specify them explicitly in your report.

No

It is a condition of publication that authors make their supporting data, code and materials available - either as supplementary material or hosted in an external repository. Please rate, if applicable, the supporting data on the following criteria.

Is it accessible?

Yes

Is it clear?

Yes

Is it adequate?

Yes

Do you have any ethical concerns with this paper?

No

Comments to the Author

I appreciate the way that the authors have addressed my concerns in this revised version.

Decision letter (RSPB-2019-2386.R1)

17-Jan-2020

Dear Dr Regoes

I am pleased to inform you that your manuscript entitled "Disentangling non-specific and specific transgenerational immune priming components in host-parasite interactions" has been accepted for publication in Proceedings B.

Open Access

Paper charges

All supplementary materials accompanying an accepted article will be treated as in their final form. They will be published alongside the paper on the journal website and posted on the online

figshare repository. Files on figshare will be made available approximately one week before the accompanying article so that the supplementary material can be attributed a unique DOI.

Sincerely,
Professor Gary Carvalho
Editor, Proceedings B
mailto: proceedingsb@royalsociety.org

Associate Editor:
Board Member: 1
Comments to Author:
(There are no comments.)

Board Member: 2
Comments to Author:
(There are no comments.)

Appendix A

Point-by-point replies to the editor and reviewers

Here we reply to each point raised by the editor and the reviewers. We copied the editors and reviewer's comments in Courier font. When we quote section from our revised manuscript, new text is shown in red.

Associate Editor

The manuscript 'Disentangling unspecific and specific transgenerational immune priming components in host-parasite interactions' was reviewed by myself and two experts in the field. As you can see from their reviews, the reviewers both thought the manuscript tested important concepts and would be of broad interest to readers.

Response: We would like to thank the editor for this encouraging assessment.

However, there are some significant points and clarifications that were raised and that should be addressed prior to publication including the following: 1) there are many writing errors throughout that make it difficult to interpret the findings. This paper needs to be carefully edited in its entirety;

Response: We apologize for the writing errors in the originally submitted version. We went through the manuscript repeatedly and carefully to eliminate orthographic and grammatical errors, especially those that make it difficult to interpret our findings. For detailed changes see our replies to the points of Reviewer 1 and 2 below.

2) both reviewers emphasize the need for more details about the modeling and model fit. The authors need to include the full modeling framework in this manuscript rather than citing a previous paper;

Response: We agree that, for the sake of self-sufficiency of our paper, previously published details of our modeling framework should be included. We have expanded the model description. However, due to page limitations of the journal, we had to move the model description into the Electronic Supplementary Material. We have also considerably shortened the description of the newly developed model in the Results, and moved the discussion of how our inference framework relates to generalized linear models to the Electronic Supplementary Material. The model is still described briefly in the Results because it is necessary to understand our findings, and because we feel strongly that the new model itself is a result, especially considering the research time that its development required.

3) the discussion should be broadened to address importance of these findings across the field and other systems as well;

Response: We would like to thank the editor and the reviewers for this suggestion. We have broadened the Discussion, and now elaborate on other systems. We have inserted an entire paragraph (lines 305–320):

Given that transgenerational immune priming is widespread, what is their adaptive significance? Two (not necessarily mutually exclusive) evolutionary hypotheses have been proposed (Roth et al., 2018). First, the transfer of immunity to an offspring may protect it during an otherwise vulnerable period, when infection probability is higher and the offspring is not yet able to mount its own effective responses (i.e., age effects, reviewed in Ben-Ami (2019)). Second, if maternal experience predicts that of her offspring, transgenerational immune priming that is dependent on maternal experience has a clear adaptive benefit. However, while this may be adaptive in terms of increased offspring protection when infection probability is higher, it may also entail additional costs associated with maintaining a high degree of immune capacity, and thus constrain the offspring (e.g., Sadd and Schmid-Hempel (2009); Zanchi et al. (2011)). For example, in populations of flour beetles infected with their natural pathogen *Bacillus thuringiensis*, females that produced more offspring had lower survival benefit, suggesting a trade-off between priming response and reproduction (Khan et al., 2019). Furthermore, if maternal and offspring environments differ or cycle, transgenerational parasite pressures will fluctuate over generations, and lead to selection on both parents and offspring (Kirkpatrick and Lande, 1989).

4) the figures need to be reworked to better convey the main important findings of the paper. In addition to these issues, there is a long list of specific suggested changes the authors should address as well.

Response: We revised all the figures, and, following a suggestion of Reviewer #2, included a new one (now Figure 4) showing a summary of all estimated ID_{50} values. In the revisions of the figures, we are now adopting an overarching color and line type scheme. Also, we kept Figure 3 (now Figure S1) but now provide a hopefully better explanation of how the figure is linked to the model description. Both the comprehensive model description and the figure are now in the Electronic Supplementary Material.

Reviewer #1:

This is indeed a nice approach to tease apart specific vs. nonspecific immune priming in insects. Overall, I find the study quite relevant to the field, providing novel insights into how survival data could be interpreted using balanced use of empirical and theoretical model.

Response: We would like to thank the reviewer for this encouraging assessment.

I suggest only a few minor points to be addressed
Introduction
- Line 19: Should it not be ‘Exposure of mothers’?

Response: We rewrote that sentence for the sake of clarity. We refrained from using the phrase “exposure of mothers” because together with following “offspring”, this could be misconstrued as an effect on grand daughters.

- Line 38-40: Please note that a recent study also showed experimental evolution of strain-specific priming in lab-bred flour beetles (Khan et al. 2017, Proceedings B, doi: 10.1098/rspb.2017.1583.).

Response: Thank you very much for pointing out this paper to us. We now mention and cite Khan et al. a little further down in the Introduction.

- Lines 43-47: It is perhaps more clear to say that the homology lies in innate immune components.

Response: We clarified this by inserting “innate”.

- Lines 84-88: Introduction to vaccine seems abrupt to general readers. Can you please rephrase or put more context to it.

Response: We agree with the reviewer that more context would be necessary to make the link to frailty approaches used in epidemiology and vaccine trials. Because this aspect is not essential at the end of the Introduction, we deleted the mention of this issue and the corresponding references. Instead, we now mention leaky and all-or-none vaccines in the last paragraph of the Discussion in which we discuss how our work goes beyond the inference frameworks available in epidemiology.

Results & Discussion

- Lines 95-96: Some Daphnia sounds strange. Please change it to ‘a subset perhaps’

Response: We changed this phrase to “a subset of Daphnia” as suggested by the reviewer.

- Line 117: Can you please clarify the parameters here?

Response: We now specify which parameters we mean here in the following text and refer to the Electronic Supplementary Material that contains a much more comprehensive exposition of our modeling framework.

- Lines 120-122 ‘‘Hereby, we separated these ... heterologous and a homologous.’’ The sentence construction seems odd. Can you please rephrase?

Response: We rewrote this sentence for clarity. The entire section, into which this sentence is embedded, has been completely rewritten. We believe the exposition of the model is more concise and clear in the revised manuscript.

- Lines 180-182: Sorry if I missed it: If the potential effect does not vary between P1, P2 & P5, then how are they considering these strains are different.

Response: At this level, we do not consider the difference in these strains in the model. The logic here is that we need to test for overall priming effects first before testing these if these effects are non-specific or specific. To clarify this issue, we replaced the original wording (which was admittedly ambiguous):

They allow for non-specific and specific effects on the average susceptibility or its variation.

with the following statement (lines 169–170):

These model extensions test for the existence of overall immune priming effects of maternal exposure on the average susceptibility or its variation.

- Line 185: ‘‘We estimate an effect $r = 0.43$.’’ Can you clarify how did you fit the model and get the value of r ?

Response: In the revised paper, we now state (lines 172–173):

We fit all four model extension maximizing the likelihood function described in the Electronic Supplementary Material.

- Lines 191–192: Different parameters of the r model should be clarified.

Response: We now briefly introduce all model parameters in the Result section that describes our model and refer to the Electronic Supplementary Material for details (lines 145–148):

Our modeling framework also allows for non-specific and specific alterations of the variance of the susceptibility distribution. These effects are captured by the parameters ρ and μ , respectively (see Electronic Supplementary Material).

- Line 252–253: Could you explain clearly how trade-off between virulence and infectivity influence priming?

Response: In the revised manuscript we now elaborate on this point (lines 257–259):

If virulent strains produce fewer transmission stages, this could influence the evolution of specific immune priming and transgenerational memory.

- Can the modeling framework be a part of the materials and methods rather than result? This is because this section mostly explains the model parameters.

Response: We thank the reviewer for this suggestion. We put considerable (needed) work into restructuring the model exposition. The modeling framework is now comprehensively described in the Electronic Supplementary Material, including previously published aspects of the framework. We still keep a much shortened description of the modeling, parameters and inference in the Results because this is needed to understand our findings. We still kept a dedicated subsection describing the modeling in the Results because developing our modeling and inference framework took considerable time and represent a major methodological advance,

Methods

- Lines 335–336: ‘‘In the exposed groups, we only included the offspring of mothers that became infected upon exposure.’’ How was this ensured?

Response: We now clarify this by writing (lines 372–375):

In the exposed groups, we only included the offspring of mothers that became infected upon exposure (by collecting all offspring and using them in the second generation experiment, and later discarding from the analysis those hosts whose mothers had not been infected).

- Lines 364–365: ‘‘The experiment ended on day 44, upon which all animals were scored by eye for infection, and their infection status was recorded.’’ While in the results they mention that ‘‘After 30 days, the infection status of each offspring *Daphnia* was determined, resulting in data on the fraction of infected *Daphnia* as a function of the exposure dose.’’

Response: We thank the reviewer for spotting this mistake. We corrected and clarified the description in the Results (lines 109–112):

Lastly, we assessed the infection status of *Daphnia* offspring 39 days after exposure (at day 44). The experiment is shown diagrammatically in Figure 1. The main readout from this experiment is the fraction of infected *Daphnia* as a function of the exposure dose.

Reviewer #2:

In this manuscript, Ben-Ami and colleagues explore the effect of pathogen priming on transgenerational susceptibility using a fully-factorial experiment and mathematical modeling framework. The experiment they conducted was novel, the findings were generally exciting, and the modeling appeared robust.

Response: We would like to thank the reviewer for this generally positive assessment.

However, while I found the content generally interesting and the experiment novel, I had a difficult time deciphering the primary findings and felt that the writing needed to be substantially revised to improve clarity, reorganize sections, remove typographical errors, and improve sentence structure. I also felt the work could be improved upon through larger attention to broader theory, and a restructuring for the primary graphs and results to draw attention to the core findings of the work.

Response: We thoroughly revised our manuscript, and completely restructured and rewrote a substantial fraction of the text and figure. The detailed changes are listed below.

General comments:

Line numbers would have been helpful in providing feedback.

Response: We apologize for this oversight. We believed they would be added at the PDF conversion stage of the submission system. We provided a file with line numbers at the request of the editors, but that file might have been transmitted too late for this reviewer. We now provide line numbers in our revised manuscript.

There are a few word choices that seem at odds with other literature on similar topics. For example, the authors use ‘‘unspecific’’ throughout when the more frequently used term is ‘‘nonspecific’’.

Response: We thank the reviewer for alerting us to our unorthodox word choice. We changed to ‘‘nonspecific’’ throughout the paper (including the title).

Introduction:

The introduction could be improved by more clearly stating the broader goal of the study, and some hypotheses the authors are testing with their experiment. While I understand it would be unwieldy to make comprehensive predictions about each prime-challenge combination, a clearer statement of why the study should be conducted and how they will interpret their findings would help direct the reader toward the important results.

Response: We rewrote the introduction keeping the reviewer’s criticism in mind. In addition to the multi-dose challenge studies we present and analyze, the Introduction must contain background on immune priming in invertebrates and transgenerational effects. Juggling these three aspects is challenging, and we hope we succeeded in making the Introduction more compelling.

At the end of the Introduction, we now also state the main questions we pose with our study (lines 93–96):

The main questions we address are if the susceptibility distribution is affected by priming, and if these potential effects differ for homologous or heterologous challenges. Hereby, we do not only consider priming effects on the mean susceptibility to challenge, but also effects on the variance of susceptibilities across hosts.

Results:

Throughout the results, there are multiple typos and several instances where plural is used when the term should be singular, and vice versa. In addition, many sentences are presented in a choppy manner such that it is difficult to discern the object of subsequent sentences, and the manuscript was much longer than was strictly necessary for understanding the findings.

Response: We went thoroughly through the manuscript, and hope to have improved the writing and eliminated the ambiguities that existed in the previous version. We added a file to this submission that contains the manuscript with all the changes are marked in red.

2.1 Dose dependence

A few specific examples of cases where the writing could be improved:

The authors state:

After 30 days, the infection status of each offspring *Daphnia* was determined, resulting in data on the fraction of infected *Daphnia* as a function of the exposure dose. The experiment is sketched in Figure 1. Could be written as: We assessed the infection status of *Daphnia* offspring after 30 days and determined

the fraction of infected *Daphnia* as a function of exposure dose. It would be helpful to move the call out to Figure 1 following the "To determine the existence...", the first sentence of the paragraph. The word "sketch" is informal, and you can just refer to the figure in parentheses after your first sentence about the experiment.

Response: We thank the reviewer for this suggestion, which we follow in the revised manuscript. Because the paragraph was written in the passive voice, we rewrote the rest of the paragraph also.

Moreover, we refrain from using the word "sketch" as suggested. We were not aware that the word "sketch" is informal, we thought it is a word usable in formal communication describing an informal drawing. We still kept a separate sentence for the reference to the figure because it diagrammatically represents the entire paragraph, not just to the last sentence.

We now write:

Lastly, we assessed the infection status of *Daphnia* offspring 39 days after exposure (at day 44). The experiment is shown diagrammatically in Figure 1. The main readout from these experiments is the fraction of infected *Daphnia* as a function of the exposure dose.

Another example:

"Offspring from mothers that were but did not become infected were not used." Could be written as: "We excluded offspring from mothers that did not become infected during the experiment."
"

Response: We revised the manuscript according to the reviewer's suggestion.

In general, these paragraphs needed to substantially revised and condensed.

Response: We substantially revised these paragraphs as requested. However, we did not delete or shorten the outline of the experiments that they contained because it is necessary to follow our findings, especially with the Material and Methods being at the end of the manuscript.

2.2 Modeling framework

The authors need to include the full modeling framework and additional information about how the models were fit in the methods. It is not acceptable to refer a reader to a previous manuscript for all the details.

Response: We thank the reviewer for this comment. Issues with the exposition of the modeling framework have also been raised by the editor and reviewer #1. In the revised manuscript, we have substantially expanded and revised the model description. A comprehensive description — including an overview of our previously published inference framework, on which our current study is based — can now be found in the Electronic Supplementary Material. In the Results the modeling framework is described only to the extent that is necessary to follow the paper.

Referring to the modeling framework as capturing all "conceivable" ways of how exposure could alter susceptibility is a little strange. I would recommend revising this. Again, choppy sentences make the description a bit difficult to understand and there are some awkward switches between active and passive voice.

Response: The model framework description is substantially revised and we now refrain from stating that it captures all conceivable ways of priming effects. This oversight was due to a lack of imagination when we wrote our original manuscript.

In the model description, there is no explanation of the parameters following the description of the model. This needs to be added. It is also unclear whether each b here is an individual *Daphnia*, and the subscripts need to be described.

Response: The model parameters and subscripts are now introduced partially in the Results and comprehensively in the Electronic Supplementary Material.

From ‘‘The above modeling framework...’’ to end of the section - it is unclear why any of this information is in the results section, particularly in the context of all the missing information about the model. This seems like a response to reviewers that is oddly placed in the results section but contains no references, and I had no idea why the authors were explaining this. This would be better suited to another paper or explained in the supplemental material.

Response: The reviewer is correct that this is a reaction to a point a previous reviewer raised. We also realize that the inclusion of this material should have been better motivated.

We feel very strongly that the relationship of our statistical inference scheme to generalized linear models is an important point that is central to our study. A previous submission of our study was rejected by *PLOS Pathogens* because one of the reviewers and the editor did not see why a mathematical model would be needed to disentangle specific from non-specific effects. Apparently, the community is using generalized linear models — very often without giving details that would allow readers to assess the validity of the analysis.

We therefore chose to briefly mention the relationship in the end of the model framework section of the Results, and elaborate on this relationship in the Discussion. Following the reviewers suggestion, we moved the detailed comparison into the Electronic Supplementary Material.

The modeling description is insufficient to understand how the authors estimated parameters or fit the model. I would recommend adding all the details to the Methods section with a brief introductory description here.

Response: We completely rewrote the section describing the model. The model is now described in the Electronic Supplementary Material, including its basis in previous work. In the Results, we now summarize the newly developed model only to the extent required to understand the detailed results.

I found Figure 3 to be confusing and was not sure that it helped with interpretation. I do not think this needs to be included as a main figure, or it needs to be substantially revised to explain how it helps to assess whether immunity is specific or non-specific.

Response: We are very grateful to the reviewer for pointing out that the figure could not be understood. Figure 3 is central to explaining our modeling approach. We therefore have kept it but moved it into the Electronic Supplementary Material with the model description. We hope that the figure can now be understood when combined with the more comprehensive modeling description, and the effort we put into improving the explanation of the figure by linking it to the relevant aspect of the model (Equation 9).

2.3 Baseline susceptibility It is unclear why information gained on the estimates of average and baseline susceptibility is important and therefore why this is being described. The last paragraph of this section needs to be revised for grammar and writing.

Response: To clarify the point of this section, we inserted as the second sentence:

The baseline susceptibility estimated are an important reference point against which we later tested for non-specific and specific immune priming effects.

In the previous version, there was already such a statement, but more towards the end of the first paragraph and might have gone unnoticed.

We also rewrote the last paragraph as suggested. It now reads:

To study if there is non-specific or specific transgenerational immune priming, we adopted a model selection scheme. We constructed models with increasing complexity, the simplest of which assumes no immune priming effects, and serves as a null model in our analysis. Table 1 lists and defines the models we considered. By fitting the models in order of increasing complexity to our experimental data and comparing the quality of the fits statistically, we test for the existence of non-specific and specific immune priming effects (see Electronic Supplementary Material).

2.4 Evidence for.. "There are four" - Should be "We tested four possible models"

Response: We followed the suggestion of the reviewer and changed the text accordingly.

"While all model extensions".. This is a central result but is lost by having to comparing ID50 values between 5A and 5B. It would be better to include a summary graph showing the ID50s and various fits, or some other summary figure that could better highlight this result.

Response: We thank the reviewer for this suggestion. We produced a figure showing the ID₅₀ values, which is now Figure 4 in the revised manuscript.

We include this new figure below for convenience:

2.4 & 2.5 - These headers seem to be indicating identical results

Response: Section 2.4 is entitled “Evidence for unspecific immune priming”, and the title of Section 2.5 is “No evidence for specific immune priming”. In our view, these are clearly different findings. The Reviewer seems to implicitly assume some relation between non-specific and specific immune priming that would make these two findings equivalent. We are not aware of such a relation. Other conceivable findings would be:

- evidence for unspecific immune priming and evidence for specific immune priming, or
- no evidence for either, non-specific or specific immune priming.

We hence decided to leave the headers unchanged.

2.6. last paragraph: The way these parameters are described makes it very difficult to understand what they are indicating. They seem to suggest facilitation of infection for the P5 strain, but this was not clear.

Response: We rewrote this last section to clarify what the parameters mean. We hope that, combined with the more comprehensive description of the model, this preempts confusion.

Specifically, we explain the last two most relevant models in more detail. In the second paragraph of this section, we now write (lines 195–205):

The most relevant extensions are the $r - m_i$ -model and the $r_i - m$ -model. The $r - m_i$ -model assumes that maternal exposure to any of the three parasite isolates reduces the susceptibility of the offspring non-specifically by the same fraction r . The model further assumes three specific effects, measured by m_i , describing how maternal exposure to each parasite isolate reduces the susceptibility of the offspring to homologous challenge with the same parasite isolate. The $r_i - m$ -model, in contrast, assumes that the non-specific reduction of the susceptibility of offspring differs for each of the parasite isolates, to which the mothers were exposed. The specific effect, on the other hand, is assumed to be the same for each parasite isolate, i. e. the offspring’s susceptibility to homologous challenge is assumed to be reduced by the same fraction m for each parasite isolate.

And in the last paragraph we now added (lines 212–215):

The non-specific effect of maternal exposure in the $r - m_i$ -model is estimated as $r = 0.48$ with a 95% confidence interval between 0.39 and 0.55. This means that maternal exposure to a parasite isolate reduces the susceptibility of the offspring to any of the three parasite isolates by 48%.

Discussion:

In general, I felt the discussion was fairly narrow and lacked a broader view of what the results indicated about transgenerational priming as it might apply to other systems. Also, the writing needs to be carefully revised to be clearer and ensure all sentences are complete.

Response: We carefully revised the writing in the Discussion (as we did for the entire manuscript).

To discuss systems beyond *Daphnia*, we added a new paragraph to the Discussion and added some material to existing paragraphs. We paste the major changes here for convenience (lines 295–320):

Transgenerational immune priming has been described in a variety of taxa, including insects and crustaceans (reviewed in Roth et al. (2018)). While most studies could not disentangle non-specific from specific immune priming by design because the mother and the offspring were exposed to the same parasite strain, there is mounting evidence of specific immune priming in invertebrates (Khan et al., 2017; Little et al., 2003; Norouzitallab et al., 2016; Roth et al., 2010). Here, we present the most extensive data set and analysis of transgenerational **specific** immune priming in invertebrates to-date. While we find evidence for specific priming, albeit in the “wrong” direction of facilitation, rather than protection, by **one of the three parasite strains** (P5), we do not find any support for the idea of a specific priming effect (or “memory”) that applies to more than one of parasite strains in the *Daphnia-Pasteuria* system.

Given that transgenerational immune priming is widespread, what is their adaptive significance? Two (not necessarily mutually exclusive) evolutionary hypotheses have been proposed (Roth et al., 2018). First, the transfer of immunity to an offspring may protect it during an otherwise vulnerable period, when infection probability is higher and the offspring is not yet able to mount its own effective responses (i.e., age effects, reviewed in Ben-Ami (2019)). Second, if maternal experience predicts that of her offspring, transgenerational immune priming that is dependent on maternal experience has a clear adaptive benefit. However, while this may be adaptive in terms of increased offspring protection when infection probability is higher, it may also entail additional costs associated with maintaining a high degree of immune capacity, and thus constrain the offspring (e.g., Sadd and Schmid-Hempel (2009); Zanchi et al. (2011)). For example, in populations of flour beetles infected with their natural pathogen *Bacillus thuringiensis*, females that produced more offspring had lower survival benefit, suggesting a trade-off between priming response and reproduction (Khan et al., 2019). Furthermore, if maternal and offspring environments differ or cycle, transgenerational parasite pressures will fluctuate

over generations, and lead to selection on both parents and offspring (Kirkpatrick and Lande, 1989).

"The conflict between our study" ..to the end of the paragraph
- This was an odd ending to this paragraph, and I did not think it was necessary.

Response: The reviewer views our paper from the perspective of an expert, who is well aware of these past discussions about the existence of immune memory in invertebrates. We believe that moving from the myopic comparison of Little et al and our study to the big question could be insightful for the general readership of Proc Roy Soc B. Nevertheless, for the sake of brevity, we decided to delete these sentences.

"In summary," - this section to the end is the strongest part of the discussion. I think integrating this broader more theoretical context into the specific findings would improve the discussion.

Response: We are very glad that this last paragraph resonates with the reviewer, especially because it highlights the theoretical aspects of our work, rather than the specific biological insights we gained.

We have expanded this paragraph by elaborating on the frailty models that are used in mathematical epidemiology, and that we extended with our work. We also worked this theoretical perspective into all parts of the paper. In the Introduction, we now more clearly state our leading questions that originate from this theoretical perspective. The Results present the modeling and the theory in a more streamlined fashion. The modeling framework itself is now comprehensively presented in the Electronic Supplementary Material. We hope that these changes bring the theoretical aspects of our work more to the forefront when we talk about our specific findings.

Methods In general, the authors switch between active "We did this" voice and passive "The food levels were this" voice. This should be revised.

Response: We revised the text and are now using the active voice as consistently as possible.

"Offspring died due to injury inflicted"... This is awkwardly written and unclear why this happened. In general, the writing of this paragraph should be significantly improved.

Response: We rephrased this statement and revised the entire paragraph. It now reads (lines 399–406):

When offspring individuals died, we recorded the day of their death. The main cause of death of offspring was injuries inflicted when we separated them from their mothers. We assessed individuals that died for infection only if their death occurred more than 16 days after their birth because infection cannot be reliably determined earlier. Animals that died earlier were excluded from the analysis. We ended the experiment on day 44, and scored all animals by eye for infection. In cases, in which we could not unambiguously determine the infection status by eye, we dissected the animal and checked for infection using a phase contrast microscope (300-600X).

Figure 1. This is a very helpful conceptual figure. I really liked this diagram.

Response: We thank the reviewer for these positive comments. This figure establishes a line type and color scheme that we are now applying to all the figures of the paper: the treatment of the mother generation is denoted by different line types (control=solid, P1=dashed, P2=dotted, P5=dash-dotted), and the treatment of the offspring is differently colored (P1=green, P2=blue, P5=red).

Figure 2. The line thickness made this very difficult to discern. I would recommend changing to color instead of line thickness. Also, the abbreviations are unnecessary and the description can be written in full. I also was uncertain why any of these were displayed and what quantitative result was supposed to be obtained from this figure. The authors state that more formal analysis is needed to understand these, so why are they being shown in the main text at all?

Response: We agree that Figure 2 was crowded. To improve this figure, we now plot the fraction of infected by maternal treatment — control, exposed to P1, P2, P5. In this figure, we use the same colors and line types as in Figure 1, 4 and S2. The figure shows the core data that we analyzed and is therefore essential for understanding our experimental design and readout. We strongly believe that the key data of a paper should be shown in the main text of a paper and not be moved to the supplement just because we performed further analysis on it, especially because the figure with the model fits, which also shows the data, has now been moved into the Electronic Supplementary Material.

Here is the revised figure for convenience:

Figure 3. As mentioned above, I did not understand this figure nor could I discern how the authors came up with these predictions. More information is necessary here.

Response: As we state above in the context of an earlier point of this reviewer about Figure 3, we hope that this figure can now be understood when combined with the more comprehensive modeling description, and the effort we put into improving the explanation of the Figure by linking it to the relevant aspect of the model (Equation 9).

Figure 4: Does this figure really need to be included in the main text? It doesn't provide much information in isolation (we need to refer to Table 1 to understand the models) and the authors state in the results which models were most supported.

Response: The reviewer is correct that this figure requires to look at Table 1. It also requires to read the description of our modeling and inference and of our experimental approach. In our view, this figure visualizes the hierarchy of our model selection in the most accessible and comprehensive way. The description in the text highlights mostly the statistically supported models. Table 1 lists all models and defines them but does not show their hierarchy clearly. For these reasons, we decided to keep this figure.

Figure 5. Aside from Figure 2, which the authors state does not provide clear insight into their question, this is the primary data figure in the manuscript. However, the display of these graphs makes it very difficult to compare among groups and generally understand what each of these different panels indicates. The use of green color makes it very difficult to see, and the dotted versus dashed line is unnecessary, particularly when plotted on different panels. It would also be easier to compare across groups if these were plotted on the same graph, for example combining either columns or rows. Including a separate panel of ID50s (for example as a bar graph) would also provide for insight than comparing numbers across graphs.

Response: We followed the reviewers suggestion and produced a figure with the ID₅₀ values. We added this new figure as Figure 4 (see above).

We moved Figure 5 into the Electronic Supplementary Material because of the page limitation of the journal. We revised Figure 5 for visibility by making the green darker, but kept the color and line type scheme for consistency with the other figures.

Appendix B

Point-by-point replies to the editor and reviewers

Here we reply to each point raised by the associate editor and the reviewers. We would like to thank the editor and the reviewers for the time and effort they put into assessing our paper. We believe that due to their comments our paper has substantially improved.

Below, we copied the editor's and reviewers' comments in Courier font. When we quote section from our revised manuscript, new text is shown in red. The line numbers refer to the current, revised manuscript and the Electronic Supplementary Material, which are attached to this reply letter (with changes marked).

Associate Editor:

We very much appreciate the significant improvement in your revised MS, but there remains some important suggestions from referee # 4, that I endorse. As you know, at the PRSB we always aim to publish articles that advance the field significantly with findings of broad generic interest. There is scope within your MS to enhance these elements further, with one final opportunity. While the paper does move the field forward from a methodological perspective, it does not sufficiently make clear how the results move forward our understanding of transgenerational immune priming conceptually. While the use of multiple doses in your design is commended, additional consideration of your findings from the variance / heterogeneity perspective, would be especially informative. Please note, as you will see below, it is only in exceptional circumstances do we allow multiple rounds of revision. While there is a collective view that your work has significant potential, we do require additional revision. I would hope that you are able to respond constructively and fully, since as with all peer review processes, the invitation to revise your MS does not carry with it any assumption of eventual publication. Nevertheless, we would welcome the opportunity to consider your MS one final time.

Response: We would like to thank the associate editor for these comments. We have made every effort to clarify the insights we gained into immune priming in the *Daphnia-Pasteuria* system, especially those on the heterogeneity of the offspring's susceptibility distributions.

These additions are detailed in our responses to Reviewer #4. To briefly summarize:

- We do not find evidence for changes in the variances of the susceptibility distributions due to priming. This is consistent with a leaky mode of action of priming.
- The finding of significant non-specific trans-generational immune priming together with the lack of evidence for specific priming is consistent with an evolutionary scenario in which *Daphnia* are facing a persistent pathogen environment across generations, but do not encounter the same *Pasteuria* strains from one generation to the next.
- Our study shows the limits of specificity in immune priming effects in *Daphnia*. This will inspire to test these limits also in other systems.

Reviewer #3:

Comments to the Author(s). This manuscript describes a set of fully factorial trans-generational priming experiments using three isolates of *P. ramosa* in *Daphnia*. Notably, the experiments include full dose-response estimates of susceptibility in the offspring to derive not only the mean changes in susceptibility, but also the variance in phenotypes over exposure dose. The manuscript further describes a mathematical framework that can be applied to this data to isolate interpretable estimates for the contributions of specific and non-specific priming/facilitation of a primary exposure upon the susceptibility of a host to a secondary exposure.

I really enjoyed this manuscript and think the framework ought to be a model for susceptibility and priming studies going forward. In fact, I can think of a couple of recently published specificity in priming studies that would have been further improved by this approach (there is a new and directly relevant one that the authors might be interested in citing - Ferro et al. 2019 PNAS 116(41)). I thought the discussion about the advantages of this modeling framework over GLMs was particularly thoughtful, and should be appealing to those who might not otherwise appreciate the advantages of this approach.

Response: We would like to thank the reviewer for this encouraging assessment.

I think the ESM plays a role of outsized importance in getting the big picture across here, but I also don't really have suggestions about how to better integrate some of the details into the main manuscript without significantly altering the length. The main text itself does a good job of pointing readers to the ESM, so hopefully it will be read just as thoroughly as the main text.

Response: Despite its importance, we indeed decided to place the comparison with more common generalized linear modelling approaches into the Electronic Supporting Material to comply with the space constraints of Proceedings B. We are glad to hear that we successfully point the reader to this material.

I think the manuscript is well written and clearly communicates the key concepts. I have no suggestions for revisions that would significantly improve the manuscript at this point.

Response: We are glad that the reviewer finds our expositions clear. We have carefully revised the manuscript a last time, mostly in line with the points raised by Reviewer #4 (see below).

Reviewer #4:

Comments to the Author(s). This manuscript examines specific versus non-specific transgenerational immune priming in a *Daphnia*-bacterial parasite system, using distinct parasite strains. This is clearly

a very well done and thorough study, which uses a novel and powerful modeling framework with strong broader utility well beyond this system. While the methodological advances of the paper are clear from reading it, I found that the discussion did not clearly elucidate the main take-home points of the study for our understanding of transgenerational immune priming more broadly.

Response: We carefully revised the summary of the biological results of our study with regard to the *Daphnia-Pasteuria* system, and host-parasite systems in general. We have clarified and added material about the conclusions we can draw from our study for transgenerational immune priming in *Daphnia* and other systems. Our changes are detailed in response to the specific points of this reviewer below.

The lack of evidence for specific immune memory is interesting, especially given prior studies showing its presence (in one case even on the same system). But what can we conclude from this? It is interesting that most of the invertebrate studies testing for specific immunity focus on strain-level specificity of immune priming, but perhaps there is specificity present more frequently at higher levels of biological organization (bacterial species, if one can even define such a thing!).

Response: The lack of evidence for specific immune priming in our study is not simply a contradiction to the study by Little et al in the same system. The apparent discrepancy stems from the fact that Little et al focused on the effects of maternal exposure on the fertility of the offspring, while we investigated the offspring's susceptibility. This is detailed in the Discussion (lines 222–225):

We also added a few sentences about the limits of specificity that our study shows, and that would be important to work out also in other systems because they may help elucidating the molecular mechanisms responsible for transgenerational immune priming (see below).

I found the use of distinct doses for the challenge infection really interesting (as well as the lack of detectable effects of priming on variance), but the authors never really highlight anything about these results in the discussion (other than mentioning that their models allow for looking at heterogeneity). Thus, I was left wanting a bit in terms of what we can conclude about this field more generally after such a thorough and well-done study. The “facilitation” result was also very strange and difficult to explain in light of prior results. However, I know that ecology is messy, and sometimes there is no good explanation.

Response: In the revised manuscript, we now clearly state that we do not find an effect on the variance of the susceptibility in the offspring population, although our method could have detected such effect on the variance. The discrepancy with our own previous work — despite our earnest efforts to understand them — remain unresolved.

Below are my specific comments, which I hope will be helpful for any revisions:

Line 19-20: awkwardly worded - can you say “with multiple doses of the same (homologous) or a different (heterologous) strain.”

Response: We rewrote this sentence as the reviewer suggests.

Line 23: ‘‘experimental assessment of vaccines’’ seems a bit strong - I might clarify that you are talking about your modeling framework here (the experimental assessment wording implies otherwise, in my opinion).

Response: We thank the reviewer for pointing out that our wording was misleading. Yes, we were referring to the relevance of our modelling approach for the experimental assessment of vaccines. We rephrased this sentence which now reads (lines 21):

Methodologically, our work represents an important contribution not only to the analysis of immune priming in ecological systems, but also to the experimental assessment of vaccines.

Lines 108: I think you should refer to supplemental materials here when you mention that you excluded offspring from moms that did not become infected.

Response: As the reviewer suggests, we now refer to the Electronic Supplementary Material in this context.

Lines 110-111: rather than saying the experiment is diagrammed in Figure 1, just cite Fig 1 after one of the prior sentences.

Response: We now refer to Figure 1 after the sentence ‘‘We then challenged the offspring individuals from the four treatment groups with seven different doses of each of the three parasite strains’’ in line 100.

Lines 113-114: similar to the above, just cite Figure 2 at the end of the prior sentence rather than having a sentence indicating what the figure shows.

Response: We edited the reference to Figure 2 according to the reviewer’s suggestion.

Lines 125-126: awkward wording. Perhaps replace with ‘‘contrast the susceptibilities of Daphnia whose mothers had or had not been exposed to this parasite strain.’’

Response: The wording was indeed awkward. We would like to thank the reviewer for their more elegant suggestion, which we now use.

Line 171: ‘‘strain’’ should be plural here

Response: We corrected this error (in line 170).

Lines 210-211: again, just cite the figure in the earlier sentence

Response: In this case, we kept the separate sentence because the figure does not support or illustrate the statement made in the previous sentence. We believe that syntactic references to figures can be more specific and thus improve clarity.

Line 220: simply say ‘‘(Figure 4)’’.

Response: While we agree that an unspecific figure reference would be shorter, we believe there is value to pointing the reader to the specific aspect of the figure that underpins our statement.

Line 233: specify that this was in the same system, as that is a critical detail

Response: Thank you very much for this suggestion. We now write:

In a previous study **on the same host-parasite system**, [45] found evidence for strain-specific immune priming.

Lines 264–266: this all seems unnecessary to write out { why not just say the later part ‘‘in the present study mean susceptibility of the control group was almost fourfold lower than previously...’’ rather than writing out all the estimates from each group, which is tedious to read.

Response: We summarized our previous results as the reviewer suggests.

Lines 271–273: awkward sentence. Again, I don’t think you need to give actual variance values – just give the percentage decrease.

Response: We simplified and clarified the wording. The revised version now reads (lines 252–255):

While the susceptibility variance of the control group between the present and the previous studies are consistent, in the present study exposure to isolate P5 did not lead to the significant increase in the susceptibility variance we found previously.

Line 304: this paragraph ends with no conclusion { what can we conclude from your study and how does that change our understanding of specific immune priming in *Daphnia*, or more broadly?

Response: We rewrote this paragraph, tying in the results of our study and added a conclusion that briefly discusses the broader insights we gained from our study for *Daphnia* and other systems (lines 275–290):

Transgenerational immune priming has been described in a variety of taxa, including insects and crustaceans (reviewed in [56]). Our work adds to the growing literature on trans-generational immune priming in invertebrates. While most studies could not disentangle non-specific from specific immune priming by design because the mother and the offspring were exposed to the same parasite strain, there is mounting evidence of specific immune priming in invertebrates [45, 33, 44, 43]. In this study, we present the most extensive data set and analysis of transgenerational specific immune priming in invertebrates to-date. While we find clear evidence for non-specific immune priming across generations, our results on specific immune priming are basically negative. Our evidence for a specific priming effect applies only to one of the three *Pasteria ramosa* isolates (P5) and goes into the ‘‘wrong’’ direction of facilitation, rather than protection. Our study thus shows the limits of specificity

of immune priming in *Daphnia*. According to our findings, *Daphnia* do not inherit a memory of the specific *Pasteuria ramosa* isolate to which they were exposed. Our study emphasizes that there are limits of specificity, even in systems where specific immune priming effects have been established. Determining these limits can contribute to identifying the often elusive molecular mechanisms that confer specific priming effects in invertebrates.

Lines 305–320: while this is important (and I think added in response to prior reviewers), it's not explicitly connected at all to your study and thus it sounds very out of place. Can you link it to what you found regarding the presence of nonspecific but not specific immune priming? In what cases would one be more beneficial than the other?

Response: We agree that linking this paragraph to our findings is desirable. We rewrote this paragraph, making it more concise and linking it explicitly to the findings of our study. The paragraph now reads (lines 291–303):

The fact that transgenerational immune priming is widespread suggests that this trait has adaptive value. Two evolutionary hypotheses have been proposed [56]. First, the transfer of immunity to offspring may protect it when it is not yet able to mount its own effective responses. This hypothesis essentially focuses on trade-off between different life stages (reviewed in [64]). Because we exposed the offspring generation early in life, this hypothesis could, at least in part, be behind the priming effect we have established. Second, if the maternal pathogen environment resembles that of the offspring, immune priming is evolutionary advantageous. Hereby, the exact degree of specificity that is most adaptive depends on how likely it is that mother and offspring are exposed to the same type or strain of pathogen [65, 66]. The non-specific priming effects we have found could have evolved in response to persistent pathogen pressure across generations. The fact that we could not find evidence for specific immune priming is consistent with an evolutionary scenario, in which subsequent generations face pressure from various types pathogens rather, than the same strain of a pathogen, such as *Pasteuria ramosa*.

Lines 321–338: I like that you discuss where your modeling framework came from but you don't discuss any of the heterogeneity results of your study, despite the unique use of multiple doses. Is it surprising that you didn't find any change in variance with priming? Does this shed light on whether transgenerational immune priming acts akin to all-or-none versus leaky vaccine effects? I assume leaky given that there was no variance effect, but I haven't thought very deeply about this and you all have. It just seems strange that you take the time to describe these hypotheses but don't relate them at all to your own study. And given the use of the varying doses, which is quite unique, it would be nice to highlight some of the inference you can gain from that aspect of the study.

Response: We thank the reviewer for raising this point. We indeed did not find evidence for a change in variance due to maternal exposure. The reviewer is correct that this means

that the effects we find are leaky rather than all-or-none. In the revised manuscript, we clarify this issue by adding the following sentences to this paragraph (lines 319–322):

In this study, we provide such an extension. Furthermore, the lack of evidence for priming effects on the variance parameters of the susceptibility distributions strongly suggest a predominantly leaky mode of action of the non-specific trans-generational priming on susceptibility.

Line 341: this seems like an overstatement. Generalized linear models can certainly distinguish between homologous and heterologous challenges - though not as well as your method.

Response: It is true that generalized linear models can be tweaked to deal with homologous versus heterologous challenges. As we outline in the Electronic Supplementray Material, this often happens at the cost of introducing a separate categorical variable that is not statistically independent of the variables capturing the maternal and offspring challenge strain. We have therefore qualified our statement (lines 325):

Most importantly, they need to be tweaked to distinguish between homologous and heterologous challenges.

Line 356: this may be a naïve question but how is transgenerational immune priming relevant in a castrating parasite? I assume the offspring are produced just prior to castration?

Response: This is a legitimate question. Actually in *Daphnia* castration is reversible, as *P. ramosa* does not destroy reproduction organs, such as, for example, trematodes, which destroy the gonads of gastropods. Instead, *P. ramosa* causes the *Daphnia* host to stop reproducing, but some hosts exhibit a burst of reproduction prior to death, known as castration relief (Mageroy et al. 2001, Parasitology; Clerc et al. 2015, Proceedings B; Ebert et al. 2016, Advances in Parasitology).

We added the following clarification to the revised manuscript (lines 339–342):

This bacterial parasite castrates its host and has a strictly horizontal transmission strategy, by releasing spores from the cadaver of infected *Daphnia* that are ingested by susceptible *Daphnia* [75, 76]. The castration, however, is not immediate and hosts can exhibit a burst of reproduction prior to death [77, 78, 76].

Line 404: a bit more details on how you score by eye would be helpful.

Response: We added the following description to this sentence (lines 387–389):

We ended the experiment on day 44, and scored all animals by eye for infection by examining the color of infected animals, which lose their typical transparency and turn brownish-red, and also lack eggs.

Table 1. Can you bold the model that you found the strongest evidence for, to make it easy for the reader?

Response: This is a good suggestion. The best model is now highlighted.

Fig 2: I agree with prior reviewers that this figure is not particularly useful, especially with the addition of Fig 4 (could confidence intervals of some kind be added to that figure?). If you do keep in Fig 2, I would recommend adding the control lines (unprimed mothers) to each of the other three strains for comparison and remove the first panel altogether- otherwise it is impossible to conclude anything from the graphs now that they are separate (though I understand why they were separated out for visual clarity...)

Response: The criticism of the previous reviewers was based on the visual similarity of Figure 2 and the figure with the model fits that are now Figure S3 (and were Figure S2 in the previous submission). We strongly believe that it is of value for a reader to see the data we collected in this form. Figure 4 presents the ID_{50} s. While these entities allow us to show the central insights of our study, they are the results of extensive analysis of our data with mathematical models. For this reason, we kept Figure 2.

The problem with the reviewer's suggestion to add the control data is that there is three of them that are all interesting point of comparison. It is more natural to show the fraction of infected hosts by offspring parasite. This is a figure which we presented in our original submission to Proceedings and which a previous reviewer found to crowded. To cater to all tastes, we have added this figure as Figure S2 to the Electronic Supplementary Material and refer to in the legend of Figure 2.

Fig 3: Is it possible to combine this with the Table in some way?

Response: We considered to combine Figure 3 with Table 1 before. Unfortunately, the detailed model definitions would terribly clutter the figure. We believe that the model definitions in Table 1 are essential information that is most economically communicated in a table format, both in term of brevity and the convenience of being able to look model definitions in one place.

Fig 4: as mentioned in comment above, can you add binomial confidence intervals of some kind to these?

Response: Adding standard errors to Figure 4 is unfortunately not straight-forward. Binomial confidence intervals are not possible for the ID_{50} estimates. They are calculated from the parameter estimates of our models b_{0j} , ν_{0j} and r , and m_i for each group, and not simply from infected fractions. We now give the expression we used to calculate the ID_{50} s in Equation 5 of the Electronic Supplementary Material.

We calculated standard errors for the ID_{50} s by parametric bootstrap, and added them to the figure. The bootstrap procedure is described in the Electronic Supplementary Material (lines 121–129).

For convenience, we show the figure here: